

# Next-generation predictive maintenance: leveraging blockchain and dynamic deep learning in a domain-independent system

Montdher Alabadi and  Adib Habbal

Computer Engineering Department, Faculty of Engineering, Karabuk University, Karabuk, Türkiye

## ABSTRACT

The fourth industrial revolution, often referred to as Industry 4.0, has revolutionized the manufacturing sector by integrating emerging technologies such as artificial intelligence (AI), machine and deep learning, Industrial Internet of Things (IIoT), cloud computing, cyber physical systems (CPSs) and cognitive computing, throughout the production life cycle. Predictive maintenance (PdM) emerges as a critical component, utilizing data analytic to track machine health and proactively detect machinery failures. Deep learning (DL), is pivotal in this context, offering superior accuracy in prediction through neural networks' data processing capabilities. However, DL adoption in PdM faces challenges, including continuous model updates and domain dependence. Meanwhile, centralized DL models, prevalent in PdM, pose security risks such as central points of failure and unauthorized access. To address these issues, this study presents an innovative decentralized PdM system integrating DL, blockchain, and decentralized storage based on the InterPlanetary File System (IPFS) for accurately predicting Remaining Useful Lifetime (RUL). DL handles predictive tasks, while blockchain secures data orchestration. Decentralized storage safeguards model metadata and training data for dynamic models. The system features synchronized two DL pipelines for time series data, encompassing prediction and training mechanisms. The detailed material and methods of this research shed light on the system's development and validation processes. Rigorous validation confirms the system's accuracy, performance, and security through an experimental testbed. The results demonstrate the system's dynamic updating and domain independence. Prediction model surpass state-of-the-art models in terms of the root mean squared error (RMSE) score. Blockchain-based scalability performance was tested based on smart contract gas usage, and the analysis shows efficient performance across varying input and output data scales. A comprehensive CIA analysis highlights the system's robust security features, addressing confidentiality, integrity, and availability aspects. The proposed decentralized predictive maintenance (PdM) system, which incorporates deep learning (DL), blockchain technology, and decentralized storage, has the potential to improve predictive accuracy and overcome significant security and scalability obstacles. Consequently, this system holds promising implications for the advancement of predictive maintenance in the context of Industry 4.0.

Corresponding author
Montdher Alabadi,
montdher10@gmail.com

# INTRODUCTION

The fourth industrial revolution, often referred to as Industry 4.0, marks a transformative integration of the Industrial Internet of Things (IIoT) into various sectors (*Alabadi, Habbal & Wei, 2022*). This integration ushers in an era of unparalleled efficiency, performance, and data-centric decision-making (*Namuduri et al., 2020*). Within this landscape, PdM emerges as a linchpin, leveraging data analytics to preemptively identify machinery malfunctions, enabling businesses to shift from reactive to proactive maintenance strategies (*Umair et al., 2021*; *Askar et al., 2022*). The significance of PdM is immense (*Nunes, Santos & Rocha, 2023*). By harnessing data from diverse sources, such as embedded machinery sensors, historical maintenance records, and real-time operational data, PdM provides in-depth insights into equipment health (*Namuduri et al., 2020*). This allows industries spanning manufacturing, healthcare, transportation, and more to significantly reduce downtime, ensure safety, optimize efficiency, and ultimately save substantial costs (*Nunes, Santos & Rocha, 2023*). Deep Learning (DL) plays a pivotal role in this context (*Ren et al., 2023*). Its multi-layered neural networks excel at processing vast datasets, unveiling intricate patterns, and delivering predictions with superior accuracy compared to traditional statistical models (*Turker & Tan, 2022*; *Altunay & Albayrak, 2023*). Traditional DL-based PdM solutions primarily rely on static models hosted on centralized servers to analyze incoming data and provide downtime forecasts, as shown in Fig. 1 (*Ran et al., 2019*). However, despite its potential, the practical application of PdM within the continuously evolving IIoT environment using DL presents notable challenges (*Chen et al., 2022b*). A primary concern is the dynamism intrinsic to the IIoT environment—conditions change, machinery evolves, and new variables emerge (*Chen et al., 2022b*). This dynamic setting necessitates constant model updating to maintain prediction accuracy (*Ren et al., 2023*). Traditional PdM solutions, being static, often struggle to adapt, leading to reduce model effectiveness over time (*Zhuang, Xu & Wang, 2023*). Additionally, the absence of domain-independent models capable of seamlessly handling various streams of multivariate time series data introduces rigidity, limiting the flexible application of PdM across diverse scenarios (*Mushtaq, Islam & Sohaib, 2021*).

The prevalent centralized data processing model in DL introduces a significant challenge (*Sengupta, Ruj & Bit, 2020*). While this approach can be efficient in specific scenarios, it also exposes critical security vulnerabilities. Centralized processing nodes become attractive targets for potential cyberattacks, giving rise to concerns about data integrity, confidentiality, and even the possibility of systemic failures (*Sanka et al., 2021*). The transmission of substantial volumes of sensitive data across networks exacerbates these risks, making the data susceptible to interception or malicious exploitation (*Boobalan et al., 2022*). Furthermore, as the Industrial Internet of Things (IIoT) continues its rapid expansion, centralized models struggle to scale effectively. The sheer volume of data threatens to overwhelm these systems, resulting in inefficiencies, increased latency, and

| Monitoring Level | Maintenance Policy |
| --- | --- |
| Cloud Level (Centralized Processing) | Predictive Deep Learning Algorithm |
| | Data Preprocessing |
| Device Level | Data Collection |

**Figure 1** **Traditional DL based predictive maintenance.**

the potential for data bottlenecks (*Kumar et al., 2023*). Edge computation and blockchain have recently emerged as a potent combination to address the above challenges (*Hafeez, Xu & McArdle, 2021*; *Shafay et al., 2023*). Edge computation enhances DL efficiency and reduces latency, crucial for real-time data processing, such as PdM in the Industrial Internet of Things (IIoT) (*Raeisi-Varzaneh et al., 2023*). However, handling DL's data volume in decentralized environments is a concern (*Shafay et al., 2023*). The InterPlanetary File System (IPFS) provides a solution by offering decentralized and distributed storage (*Kang, Yang & Zheng, 2022*). This pairing of edge computation and blockchain caters to efficient data processing and secure, decentralized storage, making it a promising solution for data-intensive DL applications like predictive maintenance. Given these limitations and considerations, the motivation for this research is to develop a PdM system for accurate Remaining Useful Lifetime (RUL) prediction that can adapt to dynamic environments, scale across multiple domains seamlessly, and enhance data and system security, all without being encumbered by the growing volume of data. To address these challenges and fulfill the aforementioned motivation, this research introduces an innovative decentralized paradigm for RUL prediction, harnessing blockchain, DL, and decentralized storage through IPFS. The proposed system is structured into three distinct levels: Device, Edge, and Monitoring levels. It leverages the combined power of blockchain technology, decentralized storage, and DL to establish a robust and efficient DL-based PdM system. The primary objectives of this research include enabling dynamic model updates, accommodating domain-specific intricacies and data requirements, and partially decentralizing control entities. The system's core design revolves around two streamlined DL pipelines: the prediction pipeline for RUL estimation and the training pipeline for continuous domain adaptation. This study's contributions can be summarized as follows:

1. Introducing a dynamic, decentralized PdM system for accurate RUL prediction by harnessing blockchain, decentralized storage, and dual DL pipelines.
2. Crafting three distinct smart contracts tailored for node registration and authentication, DL model configuration management, and the publication of newly trained model addresses within the decentralized network.

3. A simulation based testbed has been developed for validation and assessment of the proposed system using a benchmark dataset named (N-CMAPSS), emphasizing its capability in the realm of PdM.

This article follows a specific organizational structure. 'Related Work' offers a detailed examination of the relevant literature, situating the research within the current academic framework. The novel approach is expounded upon in 'Proposed System Architecture', providing a comprehensive analysis of the fundamental elements and principles of the decentralized PdM system. 'System Pipelines' will focus on the examination and analysis of system pipelines and algorithms. 'Materials and Methods' of this research paper is devoted to providing a thorough and detailed account of the experimental setup, encompassing both the materials used and the methodology employed. In 'Results and Discussion', the results are presented, providing empirical validation and performance analysis. Additionally, the significant findings are summarized, and potential avenues for future research in this field are outlined. Ultimately, the research concludes with a final conclusion.

## RELATED WORK

In this section, this research explores previously conducted work in the realm of predictive maintenance. While many studies in the literature address PdM using diverse techniques, this study focuses on those that implement ML and DL. ML and DL have driven revolutionary transformations in industrial processes over the last decade (*Zhang & Chen, 2020*). The data-rich environments of contemporary industries offer fertile ground for ML applications, leading to enhancements in industrial procedures, system dynamics, and decision support, particularly in PdM (*Mohindru, Mondal & Banka, 2020*). The study by *Feng & Li (2022)* presents an integrated decision model for manufacturing system concurrent production and maintenance decisions. This method combines a Markov chain for system analysis, a neural network for making decisions about dynamic maintenance, and an event-based detection method for finding bottlenecks in machines that need to stop. It proposes a neural network approach to determine the optimal maintenance policy for each machine in the production system, considering the current state of the system, machine health status, buffer levels, and maintenance decisions. A reinforcement learning algorithm aids in real-time decision-making. The proposed paradigm mandates continuous health monitoring for each machine, posing challenges, especially in expansive systems. The model presupposes known deterioration modes for each machine, a presumption that might not align with real-world scenarios. Additionally, it does not offer domain independence and lacks provisions for data security.

*Li et al. (2022)* unveils a novel framework for time-series production forecasting, merging the prowess of the Bidirectional Gated Recurrent Unit (Bi-GRU) network with the Sparrow Search Algorithm (SSA) for optimal hyper-parameter tuning. While the Bi-GRU network excels at decoding intricate relationships within production series, the SSA ensures proficient model hyper-parameter optimization. Despite the promising features of the framework, its evaluation is limited to three experimental trials, potentially inadequately verifying its wider applicability and effectiveness. Moreover, the paper omits

discussions on potential limitations or concerns, especially regarding security and dynamic model updates. Addressing these concerns is paramount for a holistic understanding and successful implementation of the proposed framework.

In the study by *Ong et al. (2022)*, the article delineates a comprehensive manufacturing facility for predictive equipment maintenance in the IIoT, encompassing an edge cloud, edge sensors, and human resources interconnected through a network. The proposed PdM framework encompasses AI-based decision support systems, refining resource management in both the physical and human domains. An innovative concept introduced in this study is the equipment severity rating, which quantifies the likelihood of equipment failure in comparison to health indicator values discussed in PdM literature. This method augments data-driven maintenance decision-making. Nevertheless, the study overlooks critical challenges such as data security, privacy, and the flexibility of the model to environmental shifts, as well as its capability to manage data from diverse sources. These elements are vital for a holistic, adaptable, and secure PdM system and merit further exploration.

*Bharti & McGibney (2021)* introduces SplitPred, an architecture designed for collaborative PdM. It employs local edge devices within a federated learning (FL) client's network to collaboratively train a global model. Using split-learning techniques, SplitPred facilitates edge devices to offload a segment of their model training tasks to other edge resources in the same network, optimizing resource utilization. Notably, SplitPred does better than standard horizontal cross-device FL because it ensures reliable model training at FL clients, which fixes the problems with traditional FL-based methods that don't let you share resources. Despite its advantages, the paper does not delve deeply into the framework's adaptability to variations in data scale and dimensions. Handling concurrent data streams from multiple sources also remains unaddressed. Both aspects are crucial for a flexible and dynamic PdM architecture.

*Lu & Lee (2022)* focuses on equipment maintenance, emphasizing the role of Prognostic and Health Management (PHM) in curbing maintenance costs. The study proposes the Kernel-Based Dynamic Ensemble Technique (KDET), bolstered by an Inference Confidence Index (ICI), to enable dynamic modifications and model retraining for RUL predictions. The proposed KDET system comprises three modules: offline training, weighted majority, and dynamic ensemble. The research juxtaposes KDET's performance with diverse ensemble and model retraining methods for RUL prediction, using two datasets from an accelerated deterioration experiment and the 2012 IEEE PHM Challenge. The study's findings underscore KDET's capability to dynamically amalgamate multiple models for robust RUL prediction, thereby enhancing the PdM system. However, the research doesn't address data and model security. Handling heterogeneous data from a plethora of sources also remains unexplored, underscoring vital considerations for a robust PdM framework that necessitate further investigation.

The research by *Catelani et al. (2021)* outlines a hybrid approach for predicting the RUL of lithium-ion batteries to improve operations and maintenance. This method amalgamates a condition monitoring unit with a physical deterioration model. For RUL predictions, the methodology employs state-space estimation combined with an AI estimation technique anchored on a Deep Echo State Network (DESN). The proposed single exponential model

has been demonstrated to be effective, with fewer parameter prerequisites and reduced complexity, for three out of the four tested batteries. Nonetheless, the primary limitation is the requisition of a voluminous dataset, generated through state-space estimates of recorded data, for network training. This procedure could be computationally taxing and time-consuming. Furthermore, the research omits discussions on data security, which is paramount in our data-driven era. The model's scalability to manage data from diverse sources is also unaddressed, potentially limiting its real-world applicability in heterogeneous data environments.

The study by *Wu et al. (2021)* sketches a two-tiered approach using an autoencoder-based deep neural network (AE-DNN) coupled with regression models grounded in shallow neural networks. The research introduces the concept of a degradation-aware long short-term memory autoencoder (DELTA) crafted to discern and capture varied levels of RUL deterioration trends, enhancing prediction accuracy. The DELTA framework, integrating ML-based classification with regression, has showcased superiority in RUL predictions for industrial IoT systems over existing methodologies. The article underscores the pivotal role of the AE within the DELTA framework and explicates the performance enhancements brought about by the healthy stage classification and AE components. However, the proposed methodology mandates a substantial volume of training data for optimal accuracy, and it presupposes a system deterioration model without substantial empirical evidence. Furthermore, the study overlooks vital facets like data and device security, domain-independent operation, and dynamic model updates in response to environmental fluxes. The ensemble long short-term memory neural network (ELSTMNN) is unveiled in *Cheng et al. (2021)*. This method for making RUL predictions combines predictions from different long short-term memory neural networks (LSTMNNs), each of which was trained on a different set of historical data. A novel ensemble method based on a Bayesian inference mechanism is proposed to coalesce these predictions for optimal RUL estimations. The proposed RUL prognostication methodology, validated using two discrete turbofan engine datasets, showcases its dominance over individual model-based methodologies. However, the validation methodology remains confined to these two datasets, evoking concerns regarding the method's efficiency for diverse equipment types. Moreover, the study remains silent on security considerations, which are indispensable in our data-centric age. It also does not provide insights on dynamic model updating mechanisms, potentially restricting the system's adaptability to evolving data trends and environmental transformations.

In the article by *Zonta et al. (2022)*, a PdM model deploying deep neural networks is showcased, with an emphasis on optimizing both maintenance and production schedules in the domain of smart manufacturing. By fusing data-driven techniques with physical model-based methodologies, the model aims to prognosticate the RUL of equipment. The article acknowledges the escalating tilt towards data-driven solutions using ML. Yet, it confronts challenges when managing noisy data, which impacts prediction quality. Proposed solutions include a degradation index anchored on data similarity, but the paper doesn't provide a comprehensive validation of their efficacy. While the model offers an

innovative perspective on predictive maintenance, its potential limitations when managing noisy data and navigating dynamic ecosystems are evident.

In *Chen et al. (2022a)*, a data-driven PdM strategy deploying a bidirectional long-short term memory (Bi-LSTM) model is put forth to gauge the uncertainties in RUL predictions of systems. Trialed on aero-engine health monitoring, this methodology showcases potential for slashing maintenance costs. However, the article underscores that existing methodologies often dissociate RUL predictions from maintenance decisions without taking prediction uncertainties into account. While their strategy appears promising, its conservative approach might inadvertently amplify costs. The study recommends refining this uncertainty management in RUL predictions and emphasizes that data security remains an ongoing concern.

In *de Pater, Reijns & Mitici (2022)*, a dynamic PdM scheduling model for aircraft engine fleets is presented, leveraging convolutional neural networks (CNNs) to prognosticate the RUL of turbofan engines. While the results are on par with existing CNN-based studies, they demonstrate the imperfect nature of RUL predictions, potentially causing premature maintenance alarms. This imperfection, coupled with limited maintenance slots and capacity, might even result in engine failures. The report's framework is domain-dependent, focusing on a specific dataset, and doesn't evaluate the applicability of other ML algorithms for RUL prognostication. For a more holistic and domain-independent solution, further investigation across diverse aircraft systems and components is required. Notably, the study's dynamic nature and domain-specific focus hint at broader challenges in predictive maintenance, including data security concerns. The extensive literature review on the above PdM using ML and DL methodologies underscores the existence of several as summarized in Table 1. These gaps are listed in the following points:

1. **Dynamic model updating:** A plethora of studies, such as those by *Feng & Li (2022)*; *Li et al. (2022)*, introduce innovative methods yet grapple with constant adaptability in ever-evolving industrial settings. The essence of dynamic adaptability is accentuated, especially given the mutable nature of real-world data, as elucidated by *de Pater, Reijns & Mitici (2022)*.

2. **Data security concerns:** Recurrent themes across researches, notably in *Chen et al. (2022a)*; *Ong et al. (2022)*, emphasize the significance of data security, particularly within the expanding landscape of IIoT. Centralized data models pose pronounced risks in data transfer and storage. This imperative for fortified security measures is augmented by findings from *Bharti & McGibney (2021)*, spotlighting the hazards associated with sharing failure data on susceptible platforms.

3. **Domain-specific limitations:** Research outcomes, exemplified by *de Pater, Reijns & Mitici (2022)*, often exhibit domain-specific restrictions, potentially curtailing the universal applicability of these methodologies. This domain-centric focus accentuates the necessity for more adaptable solutions suitable for diverse industrial contexts.

This research aims to address the limitations observed in PdM within the existing literature. It introduces a system explicitly engineered to adapt to dynamic scenarios, ensuring domain independence and emphasizing robust security measures. Fundamentally, This study methodology integrates decentralized blockchain technology, DL techniques,

**Table 1** Summary of related works.

| Ref | Main area/use case | Methods | Main contribution | Limitations |
|---|---|---|---|---|
| *Feng & Li (2022)* | Manufacturing system decisions | Neural Network | Integrated decision model for concurrent production | Requires continuous health monitoring; Does not handle dynamic environments |
| *Li et al. (2022)* | Time-series production forecasting | Bi-GRU | Combines Bi-GRU with SSA for hyper-parameter tuning | Limited scope of evaluation; Lacks dynamic model updating; |
| *Ong et al. (2022)* | IIoT-based PdM | AI-based | Full manufacturing facility for PdM in IIoT | Lacks data security ; Model flexibility ; |
| *Bharti & McGibney (2021)* | FL-based PdM | Split-learning | Collaborative PdM using local edge devices | Requires significant data for accurate model; Lacks data security measures; |
| *Lu & Lee (2022)* | Equipment maintenance | Kernel-Based | Dynamic integration of models for RUL prediction | Lacks data and model security; Doesn't handle heterogeneous data sources; |
| *Catelani et al. (2021)* | Lithium-ion battery RUL prediction | Deep Echo State Network | Capture various RUL deterioration trends | Requires large dataset created through state-space estimation; Lacks data security measures; |
| *Wu et al. (2021)* | RUL prediction | AE-DNN | Incorporating AE into the RUL prediction framework | Requires significant training data; Lacks dynamic model updating and data security measures; |
| *Cheng et al. (2021)* | RUL prediction for turbofan engines | LSTMNN | Combines predictions from multiple LSTMNNs | Limited efficacy on other equipment; Lacks data security; |
| *Zonta et al. (2022)* | PdM in smart manufacturing | Deep neural networks | Data-driven approach combined with physical model-based methods | Struggles with noisy data; Lacks dynamic model updating and data security measures |
| *Chen et al. (2022a)* | RUL prediction with uncertainty estimation | Bi-LSTM | Combines RUL prediction with MCR function | Tends to be conservative in approach; Lacks data security measures and dynamic model updating; |
| *de Pater, Reijns & Mitici (2022)* | PdM scheduling for aircraft engine fleets | CNN | Use alarms to trigger maintenance tasks | Domain-dependent ; Lacks data security and doesn't handle dynamic environments; |

and a steadfast dedication to continuous model updates. The integration process guarantees the adaptability of the proposed system to other domains. One notable feature of this system is its versatility in application across diverse fields, distinguishing it from conventional PdM systems. Significantly, the design of the given system enables it to scale proportionally with the expansion of data volumes. Furthermore, as environments change, given predictive models adapt smoothly, ensuring remarkable precision. Significantly, the solution suggested by this study has been enhanced with comprehensive security features in order to protect data and users from any intrusions.

# PROPOSED SYSTEM ARCHITECTURE

Our proposed architecture introduces a decentralized, blockchain-based system for secure and efficient PdM using DL models specifically tailored for RUL prediction. This system, structured across three levels: Device, Edge, and Monitoring, ensures secure communication with the blockchain and decentralized storage. The primary aim of the suggested system is to provide a dynamic model updating mechanism that is robust, efficient, and adaptable to the unique requirements of various domains and diverse data inputs. Its adaptability enables the system to assimilate and learn from evolving patterns in the input data, yielding precise and timely predictions for predictive maintenance. The system presents valuable insights into time series data by dynamically adjusting to and learning from new incoming data, ensuring effective PdM and elevated operational efficiency across a myriad of industries and applications.

Our system efficiently integrates two DL pipelines—the training and the prediction— which substantially reduces the reliance on centralized control entities. This design facilitates the emergence of a PdM mechanism capable of autonomously updating and predicting based on the processed data. Moreover, the interconnected nature of these pipelines promotes efficient data sharing. This means additional data is not required to train the dynamic model, as it can extract knowledge from the data already processed by the prediction pipeline. In the suggested system, each node is endowed with a unique class value. This identifier is pivotal for determining the node's role in the system, be it a predictor, trainer, or manager. This class value serves as a digital identifier that the smart contract utilizes to dictate how to process each node's requests. When a node issues a request, the smart contract identifies its class value and adheres to a specific set of protocols, allowing or restricting actions based on the node's designated role. The architecture of the system, including its components, is illustrated in Fig. 2. Detailed descriptions of each component follow in subsequent sections.

## Device level

In the context of predictive maintenance, the device level denotes the primary source of observation from which data is gathered and subsequently analyzed to furnish accurate RUL predictions. The proposed system boasts scalability across multiple domains, each characterized by its distinctive time series data stream. The other components of the system are crafted to function autonomously within each domain. Importantly, there is no stringent environment prerequisite stipulated for the domain's deployment. This flexibility enables the system to serve a diverse array of sectors that stand to gain from predictive maintenance, such as the oil and gas industry, healthcare, and others.

## Edge level

The edge level serves as an integral component of the system, housing nodes that work in tandem to achieve the delineated objectives. Within this proposed framework, the edge level comprises multiple virtual networks, or sets ($S$). Each set ($S$) is associated with a RUL prediction derived from time-series data specific to a particular domain and mandates the inclusion of at least one trainer class node ($T$) and one predictor class node ($P$).

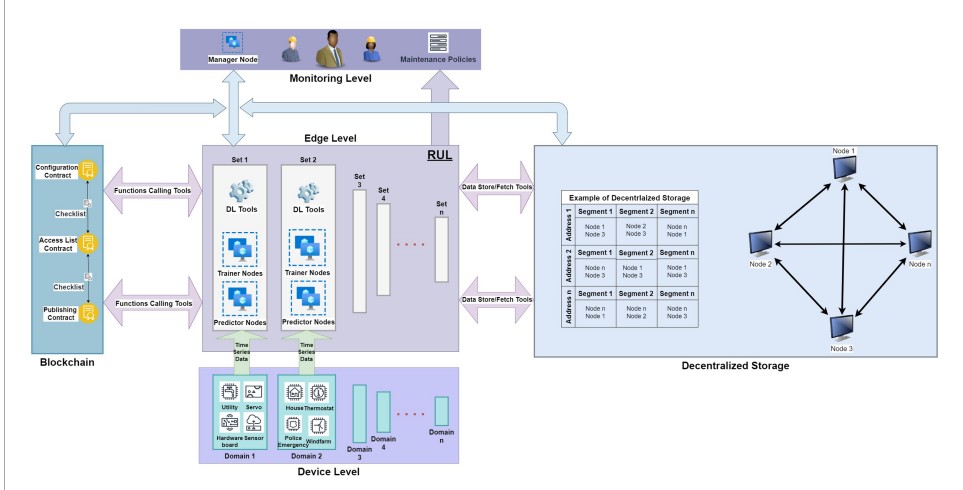

**Figure 2  Proposed system design.**

These nodes represent a diverse spectrum of devices capable of executing rudimentary DL operations at the edge level (*Raeisi-Varzaneh et al., 2023*).

Trainer nodes orchestrate continuous DL model updates, ensuring domain independence, whereas predictor nodes render RUL predictions, relying on two distinct pre-trained models: the fixed model ($Dl_e$) and the dynamically updatable model ($Dl_u$). The former is ingrained during system initiation and disseminated across the network, while the latter undergoes consistent updates. The system's RUL estimation is an amalgamation of outputs from both the static and dynamic models. This innovative approach is meticulously crafted to tackle the inherent challenges of model stability, a recurrent impediment in DL endeavors. To this end, the system employs incremental learning. Here, the dynamic model continually evolves, absorbing and adjusting to novel data, thus preserving its relevance and precision. This ensures that RUL predictions remain accurate even when confronted with evolving data patterns or atypical inputs (*Lomonaco & Maltoni, 2017*).

Governance of the nodes and models affiliated with each set falls under the purview of the system's smart contracts, a subject to be elaborated upon subsequently. Moreover, domains, along with their pertinent data corresponding to a given ($S$), are discernible through a unique identifier ($I_d$), securely ensconced within the smart contract memory. Such a configuration streamlines processes like addition, modification, retrieval, and removal from the smart contracts. To culminate, the edge-level nodes are devised to interface seamlessly with both the blockchain and decentralized storage. Both the predictor and trainer nodes come replete with the requisite resources and methodologies to navigate the system's DL pipelines. These resources and methodologies are listed in the following sub-sections.

### *Encoder-decoder DL*

Encoder-decoder DL architectures prove efficient at capturing complex temporal dependencies and patterns in data, which is crucial for time series forecasting (*Wang,*

*Su & Ding, 2021*). Long-term dependencies are common in time series data, where events that occurred in the past might have an impact on events that will occur in the future. Encoder–decoder systems are built to manage such relationships, where traditional forecasting approaches may fail to predict them *Wang, Su & Ding (2021)*. The encoder processes the input time series data and then creates a context vector of a specified size that is a compressed version of the original data. The input data's most important features and recurring patterns are summarized in this context vector. The decoder then uses the context vector to make the prediction.

Long short-term memory (LSTM) (*Hochreiter & Schmidhuber, 1997*) and gated recurrent units (GRU) (*Li et al., 2022*) are two types of recurrent neural networks (RNN) commonly used in this architecture because of their ability to deal with input sequences with long-term dependencies. The ability of these RNNs to learn and retain data from previous time steps makes them an excellent choice for time series forecasting. In this research, various combinations have been explored to implement the suggested system. These diverse combinations were systematically tested to identify the most accurate configuration, taking into consideration common hyper-parameters that significantly impact the model's performance. The proposed system ensures that only authorized users can update or delete the hyper-parameters by securely storing them within a smart contract. This approach offers multiple advantages, including enhanced security for the hyper-parameters and the model training process, leading to more trustworthy (RUL) predictions that accurately reflect the environmental observations.

## Monitoring level

The top layer is responsible for system initialization, management, and monitoring of the RUL. This level comprises the manager node ($M$), which supervises other system nodes, oversees DL initiation, and manages long-term operations. Similar to the edge level, this layer also maintains secure communication with the blockchain and decentralized storage. At this layer, system administrators have the capability to implement maintenance strategies based on the determined RUL. This study, however, will not delve into this aspect, as it predominantly centers on the system's structural framework rather than its operational intricacies. The system's design emphasizes robustness, adaptability, and efficiency, ensuring enhanced security and dependability.

## Blockchain

Blockchain is a decentralized ledger designed to enable secure peer-to-peer transactions (*Li et al., 2021*). While its initial introduction aimed to address financial issues like double-spending, blockchain's potential has expanded to bolster data security and privacy, given its strong authentication methods that involve encryption, cryptography, and immutability (*Wang et al., 2023*). A pivotal innovation in the blockchain realm is the emergence of smart contracts, which capitalize on the inherent advantages of blockchain technology (*Kumar et al., 2023*; *Karim et al., 2023*). A smart contract is a computer-coded, self-executing contract where the terms of the agreement between parties are written directly into lines of code. Stored and replicated on the blockchain, it offers a decentralized digital record (*Hu et al.,*

**Table 2   Access control contract functions.**

| Function name | Description | Authentication requirements |
|---|---|---|
| Add node | Add new node with specific class to the system | Class = M , Status = True |
| Update node | Update class and activation status of the given node | Class = M , Status = True |
| Delete node | Delete given node from the system authenticated list | Class = M , Status = True |
| Get node | Provide given node class and status | C_c ,C_p |

*2021*). These contracts automatically execute, monitor, or document events and actions of legal relevance based on predefined terms, thus obviating the need for middlemen (*Al-Amri et al., 2019*). They are architected to be secure, transparent, and resistant to alterations within a decentralized system (*Liu & Liu, 2019*).

In the proposed model, this research develops three specific smart contracts: access control ($C_a c$), configuration ($C_c$), and publishing ($C_p$) contracts. Working in tandem, these smart contracts aim to provide a domain-agnostic, dynamically updatable, and secure PdM solution. By overseeing the RUL prediction process, smart contracts augment the system's efficiency and efficacy.

### Access control contract

The access control contract, denoted as ($C_a c$), serves as the cornerstone for system access and role management. It maintains a comprehensive roster of all nodes in the system, classifying each node with its respective class designation. A node, upon its initiation, is cataloged within the memory of the smart contract alongside its associated class number. Other contracts within the system liaise with this contract to validate the legitimacy of the invoking source address. This ensures that only approved nodes have the privilege to influence the RUL prediction mechanics. In essence, ($C_a c$) acts as a gatekeeper, effectively thwarting any unauthorized external nodes from interfering in the process.

Furthermore, since each node within the system possesses a replicated copy of the smart contract, the process of assigning class numbers to individual nodes becomes pivotal. This assignment not only demarcates roles but also orchestrates permissions. Such decentralization eschews the conventional centralized oversight mechanism. Instead, the entire network democratically oversees decisions, removing the dangers associated with a single point of control or failure. Table 2 delineates the intrinsic functions embedded within the ($C_a c$) contract and outlines their pertinent execution prerequisites.

### Configuration contract

The configuration contract, represented as ($C_c$), acts as the central repository for storing the configurations—specifically, the hyper-parameters—of each DL model tied to a specific set ($S$) within the PdM system. This contract is engineered to permit transactions exclusively from manager nodes ($M$), identified by their unique identifier ($I_d$). These nodes make use of the "set" or "update" functions to either establish or modify the hyper-parameters of the DL model. After these DL hyper-parameters have been established or modified, the training nodes, labeled as ($T$), and the prediction nodes ($P$) can retrieve

---

**Algorithm 1:** Access Control Contract Algorithm

   **Data:** NodeList: Data Structure containing status,address and class

1 **Function** addNode(*Status*, *Class*, *address*):
2   NodeList[address] ← NodeList(Status, Class)
3 **Function** updateNode(*Status*, *Class*, *address*):
4   NodeList[address].nodeClass ← (Status, Class)
5 **Function** deleteNode(*address*):
6   delete NodeList[address]
7 **Function** getNode(*address*):
8   **return** NodeList[address]

---

**Table 3** Configuration contract functions.

| Function name | Description | Authentication requirements |
|---|---|---|
| Add Configuration | Add new configuration given model identification to the system | Class = M, Status = True |
| Update Configuration | Update configuration of the given model | Class = M, Status = True |
| Delete Configuration | Delete given Configuration from the system | Class = M, Status = True |
| Get Configuration | Provide given configuration | Class = P or T, Status = True |
| Change Train Status | Change the status of automatic train | Class = M, Status = True |

these parameters to incorporate them in their operational processes based on model $(I_d)$. To ensure authenticated function invocation, $(C_c)$ communicates with the access control contract $(C_ac)$. Moreover, $(C_c)$ governs the automation of DL model training *via* a Training Status $(A_t)$, which can be toggled between "true" and "false". When status is activated as true, the training process persists according to the preset protocol initialized with the system. Conversely, when the status is set to false, automatic training halts and can only be re-initiated upon request from $(M)$. Before starting the training process, trainer nodes must consistently check the status of $(A_t)$. An exhaustive description of the $(C_c)$ function is delineated in Table 3.

### *Publishing contract*

The publishing contract, denoted as $(C_p)$, is tasked with overseeing the locations of both the model and training data within the decentralized storage system. In terms of DL, this contract maintains a list presenting the address of each DL model corresponding to every individual data stream from its associated domain. This relationship is tracked using the unique identifier, $(I_d)$. At the system's outset, managerial nodes $(M)$ initiate a transaction directed at $C_p$ that conveys the storage locations of both $(Dl_e)$ and $(Dl_u)$. Furthermore, post-training of a particular $(Dl_u)$, the trainer nodes $(T)$ refresh their location by instigating a subsequent transaction directed at $C_p$. As a final point, prior to embarking on RUL predictions, predictor nodes $(P)$ consult $C_p$ to fetch the location of their linked model, encompassing both $(Dl_u)$ and $(Dl_e)$.

---

**Algorithm 2:** Configuration Contract Algorithm

**Data:** ConfigurationList:Data Structure containing model parameters

1  **Function** set Or Update Configuration($Id$, Configuration):

2  ConfigurationList[$Id$] $\leftarrow$ Configuration

3  **Function** deleteConfiguration($Id$):

4  delete ConfigurationList[$Id$]

5  **Function** changeTrainStatus($Id$):

6  ConfigurationList[$Id$].train $\leftarrow$ !ConfigurationList[$Id$].train

7  **Function** getConfiguration($Id$):

8  **return** ConfigurationList[$Id$]

9  **Function** getTrainingStatus($Id$):

10  **return** ConfigurationList[$Id$].train

---

**Table 4** Publishing contract functions.

| Function name | Description | Authentication requirements |
|---|---|---|
| Set Model | Add new model address given model identification to the system | Class = M or T, Status = True |
| Update Model | Update the address of the given model based on the given identification | Class = M or T, Status = True |
| Get Model | Provide the address of the given model given provided identification | Class = P or T, Status = True |
| Set Training Data | Add the address of the training data given associated model identification | Class = P or M, Status = True |
| Update Training Data | Update the address of training data associated with the given identification | Class = P, Status = True |
| Get Training Data | Provide the address of the training data given provided identification | Class = T, Status = True |

Shifting focus to training data management, predictor node (P) records the input data alongside their corresponding prediction outcomes, storing them as fresh training datasets. This process entails uploading the said training data to the decentralized framework, followed by a subsequent location update within ($C_p$). This ensures that the trainer nodes ($T$) can easily locate and employ this data for retraining the specified ($Dl_u$). A comprehensive breakdown of the ($C_p$) contract's operations is provided in Table 4.

## Decentralized storage

Decentralized storage emerges as a transformative and optimal paradigm, especially when interweaved with the domains of blockchain and DL methodologies (*Li et al., 2021*; *Doan et al., 2022*). Within this context, decentralized storage plays a pivotal role in accommodating

---

**Algorithm 3:** Publishing Contract Algorithm

---

   **Data:** ModelData:Data structures for storing address of Model metadata in the de-
centrlaized storage

   **Data:** TrainingData:Data structures for storing address of training data in the de-
centrlaized storage

1  **Function** setModelData($Id$, model address):

2  ModelData[$Id$] $\leftarrow$ (model address)

3  **Function** updateModelData($Id$, model address):

4  ModelData[$Id$] $\leftarrow$ (model address)

5  **Function** getModelData($Id$):

6  **return** ModelData[$Id$]

7  **Function** setTrainingData($Id$, train data address):

8  TrainingData[$Id$] $\leftarrow$ (train data address)

9  **Function** updateTrainingData($Id$, train data address):

10  TrainingData[$Id$] $\leftarrow$(train data address)

11  **Function** getTrainingData($Id$):

12  **return** TrainingData[$Id$]

---

DL metadata along with the accompanying training data. DL metadata comprises an ensemble of model-specific details, including its architectural design, associated parameters, and key performance metrics. Concurrently, training data delineates the foundational data that powers the training of these intricate models. Characterizing decentralized storage, each fragment of data is disseminated and stored across a myriad of system nodes, each piece earmarked with a distinct address to facilitate effortless access. Smart contracts, the beating heart of the blockchain, take custody of these addresses, offering a decentralized, immutable ledger that acts as a conduit to the data (*Qammar et al., 2023*). Such a structured approach to data addressing catalyzes an uninterrupted, efficient interaction, which remains paramount given the prodigious data appetites of DL paradigms.

    The onus of contributing to this decentralized storage lies heavily on the majority of system nodes, encapsulating both the edge and monitoring echelons. By propagating storage across a decentralized plane, the entire training conduit gains an efficiency boost. Data, rather than trickling from a monolithic, centralized source, cascades in parallel from divergent nodes, thereby abbreviating the traditionally elongated wait times for data accession. This symphony of parallelism also embeds redundancy into the system's DNA; with data replicas ensconced across nodes, the system's resilience to sporadic node outages is bolstered.

### IPFS

The proposed system is made up of smart contracts, an encoder–decoder DL model, and the decentralized storage power of IPFS (InterPlanetary File System). These are all carefully put together to create a safe and effective PdM environment. IPFS emerges as a peer-to-peer file dissemination system, championing the tenets of security, transparency, and robust fault tolerance in data storage and access routines (*IPFS, 2020*). The publishing contract,

denoted as ($C_p$), acts as a custodian of the addresses pointing to DL models and the encompassing training data ensconced within IPFS. This structure paves the way for fluid updates and data requisitions essential for training and subsequent predictions. The trainer nodes, represented as ($T$), are entrusted with the task of assiduously refining and training the DL models. They operate under the guidelines prescribed by the hyper-parameters domiciled within the configuration contract ($C_c$), tapping into the reservoir of training data made available *via* IPFS. Concurrently, the predictor nodes, labeled as ($P$), serve up RUL predictions predicated on pre-existing models. They consult the publishing contract ($C_p$) to glean the model addresses, subsequently reaching out to IPFS to access the requisite DL models, thereby paving the way for astute predictions.

## SYSTEM PIPELINES

The proposed system is primarily anchored around two integral DL pipelines. The first, termed the 'training pipeline', is tailored for consistent adaptation, adjusting itself to the variations evident in the input data. The succeeding pipeline, the 'prediction pipeline', undertakes the task of determining the RUL from the presented input data. This pipeline leverages insights from two distinct models: the unwavering fixed model and the adaptable dynamic model. In the suggested design, although these pipelines function autonomously, they are intricately coordinated through the system's ingrained smart contract functions, guaranteeing a smooth, synchronized operation.

Figure 3 offers a comprehensive sequence diagram that delineates the operational flow for both pipelines, encompassing the gamut of actions involved in both training and prediction phases. The visual underscores the dynamic interplay between pivotal components, such as data, models, and nodes, contributing to the successful execution of the pipelines. Complementing this, Fig. 4 vividly elucidates the inter-node interactions during the simultaneous functioning of the prediction and training pipelines. It sheds light on the symbiotic relationship between the trainer and predictor nodes, emphasizing the system's prowess in managing multifaceted operations and accentuating its robustness and adaptability in tackling real-world scenarios. Crucially, the deduced RUL in the prediction pipeline emerges from the combined intelligence of two separate models: the fixed model and the dynamic model. Such a design not only amplifies prediction accuracy but also maintains the system's evolving nature. Notably, the data from the prediction pipeline, augmented by its resultant prediction, serves as a new input for the training pipeline, fostering continuous system learning and honing its performance over time.

### Training pipeline

The training pipeline is tasked with training the DL model ($Dl_u$) associated with each set ($S$). As previously discussed, each ($S$) involves at least one trainer node ($T$). Additionally, each RUL prediction for a given ($S$) is managed by models ($Dl_e$) and ($Dl_u$), fetched using addresses stored in the publishing contract ($C_p$). The training pipeline for one ($S$) that encompasses one ($T$) and ($P$) includes the subsequent steps:

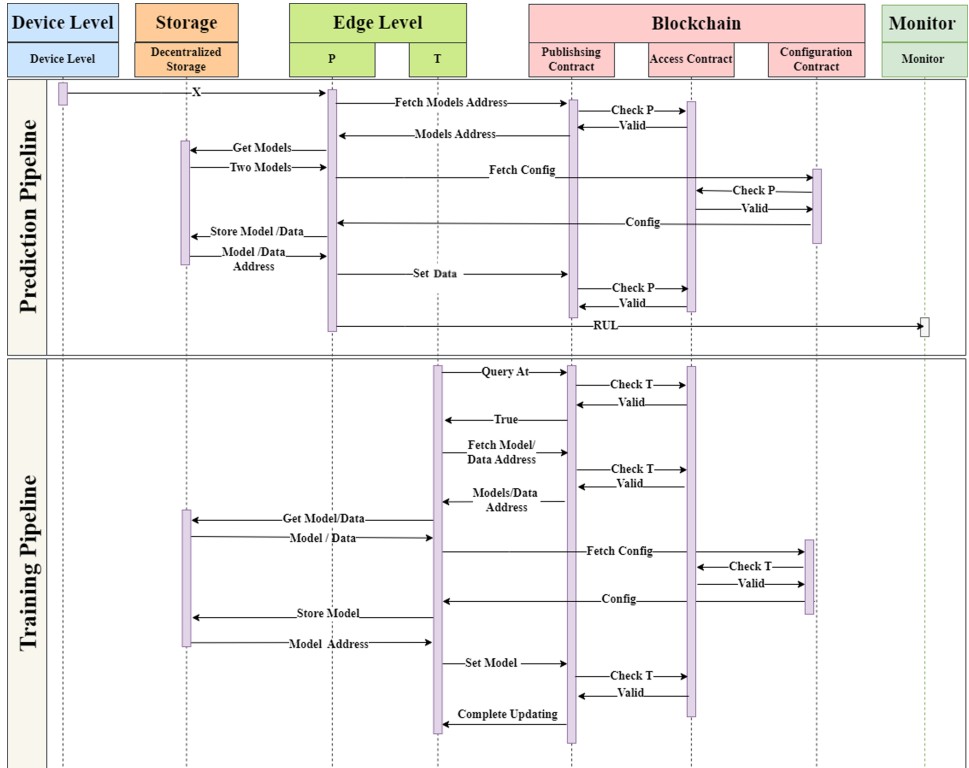

**Figure 3** Pipelines sequence diagram.

1. $(T)$ queries the configuration contract $(C_c)$ to inspect the value of the training status $(A_t)$, indicating if training is permissible for the associated set. If authorized, the execution progresses to the subsequent step. This step mandates $(C_c)$ to authenticate the status and class of $(T)$ using $(C_a c)$.

   If $(C_c \longrightarrow \text{GetNode}(C_{ac}, T)) = \text{Class(Trainer)}, \text{True}$

   $A_t = T \longrightarrow \text{GetStatus}(C_c, \text{Id})$

2. Node $(T)$ queries $(C_p)$ to fetch the address of the associated model and training data using their $(Id)$ from the decentralized storage (IPFS). This step also necessitates $(C_P)$ to authenticate the status and class of $(T)$ using $(C_a c)$.

   If $(C_p \longrightarrow \text{GetNode}(C_{ac}, T)) = \text{Class(Trainer)}, \text{True}$

   $\text{ModelAddress} = T \longrightarrow \text{GetModel}(C_p, \text{Id})$

   $\text{DataAddress} = T \longrightarrow \text{GetTrainData}(C_p, \text{Id})$

3. The model and data are fetched from decentralized storage (IPFS) utilizing the addresses acquired in the prior step.

   $Dl_u = T \longrightarrow \text{GetFromIPFS(ModelAddress)}$

   $x, y = T \longrightarrow \text{GetFromIPFS(DataAddress)}$

4. The configuration embedding the hyper-parameters for the designated model is queried from $(C_c)$ for application in subsequent steps, employing the associated $(Id)$. Also,

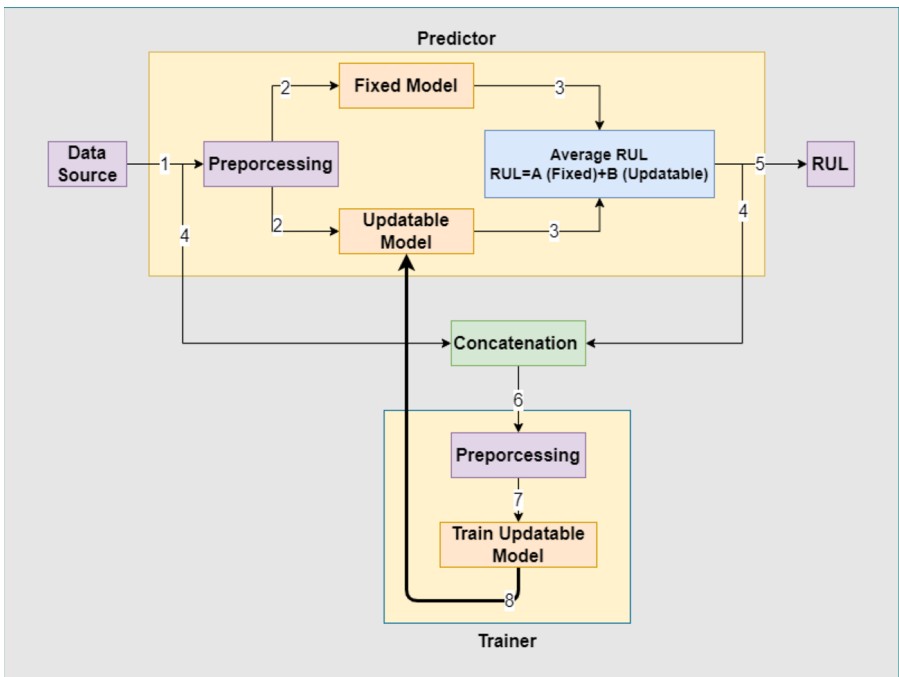

**Figure 4  Predictor and trainer collaboration diagram.**

authentication is obligatory.

If $(C_c \longrightarrow \text{GetNode}(C_a c, T)) = \text{Class(Trainer)}, \text{True}$

$\text{Config} = T \longrightarrow \text{GetConfig}(C_c, Id)$

5. The retrieved input data is processed according to the preprocessing procedure, taking into consideration the hyper-parameters, especially concerning the input data shape mandated for the encoder–decoder model.

$\text{Processed}(x, y) = \text{Preprocess}((x, y), \text{Config})$

6. Model training initiates by optimizing the loss value to amplify the predictive model's efficacy.

$\hat{y} = \text{Dl}u(x)$

$L(y, \hat{y}) = \dfrac{1}{N} \sum i = 1^N (y_i - \hat{y}_i)^2$

where: $L$ is the loss function, $y$ is the actual value, $\hat{y}$ is the predicted value, and $N$ is the count of samples.

7. The trained and updated model is stored in IPFS, and its address is refreshed in $(C_p)$ by initializing a transaction alongside its $(Id)$. Validation from $(C_p)$ is indispensable before its endorsement.

$ModelAddress = T \longrightarrow \text{StoreInIPFS}(\text{Trained}(Dl_u))$

If $(C_p \longrightarrow \text{GetNode}(C_a c, T)) = \text{Class(Trainer)}, \text{True}$ then

$T \longrightarrow \text{SetModel}(C_p, Id, ModelAddress)$

The trainer pipeline is crucial for ensuring that $(Dl_u)$ remains valid for most recent observations and can respond to any changes in input data. This process contributes to a robust and adaptable PdM system. Algorithm 4 illustrates the pipeline procedure mentioned above. It's worth mentioning that the process is similar when the case has more than one trainer node, except that the $(Dl_u)$ in the smart contract will be the last one updated in the $(C_p)$.

---

**Algorithm 4:** Training Pipeline Algorithm

---

1   T queries $C_c$ to check $A_t$
2   **if** $C_c.GetNode(C_{ac}, T) = Class(Trainer), True$ **then**
3      $A_t = T.GetStatus(C_c, Id)$
4   **end**

5   T queries $C_p$ to get model and data addresses
6   **if** $C_p.GetNode(C_{ac}, T) = Class(Trainer), True$ **then**
7      $ModelAddress = T.GetModel(C_p, Id)$
8      $DataAddress = T.GetTrainData(C_p, Id)$
9   **end**
10   $DL_u = T.GetFromIPFS(ModelAddress)$
11   $x, y = T.GetFromIPFS(DataAddress)$

12   T queries $C_c$ to get configuration
13   **if** $C_c.GetNode(C_{ac}, T) = Class(Trainer), True$ **then**
14      $Config = T.GetConfig(C_c, Id)$
15   **end**

16   $Processed(x, y) = Preprocess((x, y), Config)$
17   $\hat{y} = DL_u(x)$
18   $L(y, \hat{y}) = \frac{1}{N}\sum_{i=1}^{N}(y_i - \hat{y}_i)^2$

19   $ModelAddress = T.StoreInIPFS(Trained(DL_u))$
20   **if** $C_p.GetNode(C_{ac}, T) = Class(Trainer), True$ **then**
21      $T.SetModel(C_p, Id, ModelAddress)$
22   **end**

---

## Prediction pipeline

The prediction pipeline stands as an integral part of the system, focusing on delivering RUL predictions for time series data $(x)$. Whenever a predictor node $(P)$ receives observations from the device level, the RUL is determined through the following sequence:

1. $(P)$ initiates a query to the publishing contract $(C_p)$ to retrieve the address of the associated DL models $((Dl_e)$ and $(Dl_u))$ within the decentralized storage. Prior to releasing the necessary data, $(C_ac)$ must authenticate $(P)$.
   If $(C_p \longrightarrow GetNode(C_ac, P)) = Class(predictor), True$
   
   $ModelAddress1(Dl_e) = P \longrightarrow GetModel(C_p, Id)$

$$\text{ModelAddress2}(Dl_u) = P \longrightarrow \text{GetModel}(C_p, Id)$$

2. Next, $(P)$ extracts the relevant DL models from the decentralized storage system.

$$Dl_e = P \longrightarrow \text{GetFromIPFS}(\text{ModelAddress}1)$$

$$Dl_u = P \longrightarrow \text{GetFromIPFS}(\text{ModelAddress}2)$$

3. To fetch the configuration necessary for preprocessing, $(P)$ sends a query to $(C_c)$.

$$\text{If } (C_c \longrightarrow \text{GetNode}(C_a c, P)) = \text{Class}(predictor), \text{True}$$

$$\text{Config} = P \longrightarrow \text{GetConfig}(C_c, Id)$$

4. The incoming data $(x)$ undergoes a preprocessing procedure.

$$\text{Processed}(x) = \text{Preprocess}(x, \text{Config})$$

5. Both $(Dl_e)$ and $(Dl_u)$ are deployed to furnish RUL predictions.

$$RUL1 = Dl_e(x)$$

$$RUL2 = Dl_u(x)$$

6. Using parameters $(A)$ and $(B)$ (where $(A + B = 1)$), the final RUL is deduced *via* a weighted average.

$$RUL = (A \times RUL1) + (B \times RUL2)$$

7. Both the input data $(x)$ and the predicted RUL get stored in IPFS, updating its address in $(C_p)$ by initiating a transaction with its $(Id)$. It's pivotal that $(C_p)$ ratifies this operation prior to its endorsement.

$$\text{DataAddress} = P \longrightarrow \text{StoreInIPFS}(x, RUL)$$

$$\text{If}(C_p \longrightarrow \text{GetNode}(C_a c, P)) = \text{Class}(predictor), \text{True then}$$

$$P \longrightarrow \text{SetTrainData}(C_p, Id, \text{DataAddress})$$

In scenarios with multiple predictor nodes, the average RUL can serve as the ultimate RUL. Algorithm 5 delineates the above steps in pseudocode form.

# MATERIALS AND METHODS

## Materials

### Testbed setting

To ascertain the efficacy of the proposed system, an intricate simulation environment was established. The testbed is structured on cutting-edge hardware specifications, encompassing a 12th Gen Intel (R) Core(TM) i9-12900H processor with a clock speed of 2.90 GHz, supported by a generous 32 GB RAM, operating on the latest Windows 11 OS. System architecture incorporates multiple domains as shown before, signified as $(S)$. Each is distinguished by its unique input data but adheres to a consistent operational behavior. Initially, one set $(S)$ and one manager node $(M)$ were selected from the performance evaluations. This set, $(S)$, amalgamates distinct nodes: a trainer node $(T)$, a predictor node $(P)$. In concert, these nodes form a localized decentralized processing and storage network, facilitating efficient decentralized computations and data management.

---

**Algorithm 5:** Predicting Pipeline Algorithm

---

1    P queries $C_p$ for model addresses
2    **if** $C_p.GetNode(C_{ac}, P) = Class(predictor), True$ **then**
3        $ModelAddress1(DL_e) = P.GetModel(C_p, Id)$
4        $ModelAddress2(DL_u) = P.GetModel(C_p, Id)$
5    **end**

6    $DL_e = P.GetFromIPFS(ModelAddress1)$
7    $DL_u = P.GetFromIPFS(ModelAddress2)$

8    P queries $C_c$ for configuration
9    **if** $C_c.GetNode(C_{ac}, P) = Class(predictor), True$ **then**
10       $Config = P.GetConfig(C_c, Id)$
11    **end**

12    $Processed(x) = Preprocess((x), Config)$
13    $RUL1 = DL_e(x)$
14    $RUL2 = DL_u(x)$
15    $RUL = (A * RUL1) + (B * RUL2)$

16    $DataAddress = P.StoreInIPFS(Trained(x, RUL))$
17    **if** $C_p.GetNode(C_{ac}, P) = Class(predictor), True$ **then**
18       $P.SetTrainData(C_p, Id, DataAddress)$
19    **end**

---

For decentralized computations, the choice is to use the Ethereum blockchain (*Buterin, 2017*), predominantly using the GETH implementation, simulating an extensive blockchain environment (*Lange & Trón, 2022*). To implement a decentralized computational network, implementation instantiated a localized private GETH network. Concurrently, for decentralized storage, the IPFS client software is harnessed, ensuring a perpetual and decentralized mode of data storage and sharing. The network hosts three smart contracts: $(C_ac)$, $(C_c)$, and $(C_p)$. System nodes are then registered to $(C_ac)$ using the ensuing methods:

$M \longrightarrow Register(C_ac, (T(Class: 2, Status: True)))$

$M \longrightarrow Register(C_ac, (P(Class: 3, Status: True)))$

***Dataset***

In this research, the N-CMAPSS dataset has been used, which provides synthetic run-to-failure degradation trajectories of a fleet of turbofan engines simulated under realistic flight conditions (*Chao et al., 2021*). Specifically, the focus is on the DS02 subset of the N-CMAPSS dataset. The dataset captures measurements, observations, and conditions recorded during entire flights on a commercial jet, including various flight stages such as climb, cruise, and descent. A CMAPSS system model is employed to generate this dataset, where the inputs comprise scenario-descriptor operating conditions and latent model health parameters. The outputs include RUL estimates and other unobserved properties

not part of the condition monitoring signals, thus acting as virtual sensors. The raw dataset initially includes six categories of variables: operative conditions, measured signals, virtual sensors, engine health parameters, RUL label, and auxiliary data. Furthermore, the given dataset consists of nine engine units with numbers (2, 5, 10, 11, 14, 15, 16, 18, 20). These units are similar in terms of feature number and scale of data but differ in terms of the number of samples.

## Methods

### Pre-processing

Data pre-processing is a pivotal phase in the DL pipeline, influencing the efficacy and performance of the resulting models. In the proposed system, two pipelines leverage DL, each mandating specific data preprocessing tailored to the demands of encoder–decoder models (*Ranjan, Prusty & Jena, 2021*). The fundamental steps constituting this preprocessing procedure include:

1. **Handling missing values:** Real-world datasets frequently suffer from incomplete or absent data points. This preprocessing step rectifies the scenario either by substituting missing values with suitable replacements or excising instances characterized by data voids. Such rectification ensures algorithmic operations unhindered by data inadequacies. An initial exploration led us to impute missing data *via* the forward fill methodology, retaining the time series data's continuity.

2. **Feature selection:** Discerning and selecting salient features can augment model performance while simultaneously curtailing computational intricacy. Techniques spanning filter methods, wrapper methods, or embedded methods assist in cherry-picking the most germane features. A comprehensive exploratory data analysis, fortified by domain knowledge and precedent studies on the dataset with the correlation test, dictated the feature selection approach.

3. **Feature scaling:** Datasets often present features varying in scales. This variability can disrupt model training. Preprocessing strategies, such as normalization or standardization, harmonize feature scales, ensuring unbiased feature consideration during the model's learning phase. This was achieved this by scaling the chosen features using the MinMaxScaler technique, ensuring data normalization and outlier mitigation.

4. **Data reshaping for encoder-decoder models:** Encoder–decoder architectures demand data tailored to specific configurations, often sequential data representations. Thus, preprocessing adapts data to fit these prerequisites, which could involve transforming time series data into sequenced, fixed-length windows or modifying input data dimensions. The subsequent step involved reformatting the dataset into a three-dimensional structure, making it compliant with the encoder–decoder model prerequisites. The final dataset, post-processing, encapsulates 29 features, primed for DL models.

### Walk-forward validation

Walk-forward validation emerges as a potent tool to validate time series predictive models (*Makridakis, Spiliotis & Assimakopoulos, 2018*). It particularly shines when evaluating

models on chronologically significant data. This technique aims to mirror a model's realistic forecasting prowess, capitalizing on the latest available data. In walk-forward validation, the dataset is cleaved into sequential, non-overlapping training and validation sets. Upon an initial training bout on the training set, model validation transpires on the first test set. The training data set then ingests data from this initial validation, instigating a subsequent retraining followed by evaluation on the succeeding test set. This cyclical pattern persists until all validation sets undergo assessments.

The preeminence of walk-forward validation lies in its authentic representation of a model's real-world performance. As new data trickles in over time, the model continuously adapts, resonating with the dynamic data landscape. It also diminishes overfitting potential by iteratively testing across diverse datasets, countering any tendencies toward overoptimizing for a single dataset. Algorithm 6 offers an illustrative portrayal of this method as harnessed in suggested research.

---

**Algorithm 6:** Walk-Forward Validation for RUL Prediction

**Data:** $n_{\text{splits}}, X, n_{\text{past}}, n_{\text{future}}$
**Result:** RUL Prediction

1  $n_{\text{train}} \leftarrow \text{shape}(X)[0]/(n_{\text{splits}} + 1)$
2  $n_{\text{val}} \leftarrow n_{\text{train}}$
3  $i \leftarrow 0$
4  **while** $i < n_{\text{splits}}$ **do**
5      $train\_data \leftarrow X[: n_{\text{train}} \times (i+1)]$
6      $val\_data \leftarrow X[n_{\text{train}} \times (i+1) : n_{\text{train}} \times (i+1) + n_{\text{val}}]$
7      $test\_data \leftarrow X[n_{\text{train}} \times (i+1) + n_{\text{val}} : n_{\text{train}} \times (i+2) + n_{\text{val}}]$
8      $X\_train, y\_train \leftarrow \text{reshape\_dataset}(train\_data, n_{\text{past}}, n_{\text{future}})$
9      $X\_val, y\_val \leftarrow \text{reshape\_dataset}(val\_data, n_{\text{past}}, n_{\text{future}})$
10     $X\_test, y\_test \leftarrow \text{reshape\_dataset}(test\_data, n_{\text{past}}, n_{\text{future}})$
11     $\text{fit}(X\_train, y\_train, X\_val, y\_val)$
12     $predictions \leftarrow \text{predict}(X\_test)$
13     $i \leftarrow i+1$
14 **end**

---

### Deep learning initialization

The study integrates two distinct DL models: the $(Dl_e)$ (fixed model) and $(Dl_u)$ (dynamic updatable model). The $(Dl_e)$ model, once trained during system initialization, remains constant within the decentralized storage. It leverages dropout as a regularizing mechanism to counteract overfitting. In contrast, the $(Dl_u)$ model is inherently dynamic, warranting regular updates *via* the training pipeline. The distinction between these models is fundamentally anchored in their weight values; while the weights in the $(Dl_e)$ model remain unchanged post-initialization, the weights in $(Dl_u)$ are recurrently modified in sync with incoming data streams.

**Figure 5** Basic DL model architecture.

**Table 5** Basic setup configuration.

| Hyper-parameter | Value | Hyper-parameter | Value |
|---|---|---|---|
| Past observation steps | 25 | Dropout | 0.4 |
| Future observation Steps | 10 | Number of features | 29 |
| Validation split | 5 | A parameter | 0.9 |
| Learning rate | 0.001 | B parameter | 0.1 |

An encoder–decoder-based DL architecture underpins the proposed system, as depicted in Fig. 5. The model's hyper-parameters, inclusive of dropout layers designed to deter overfitting, are cataloged and stored within the ($C_p$) contract. A detailed exposition of these hyper-parameters is presented in Table 5. Both the ($T$) and ($P$) nodes can readily access these hyper-parameters, which facilitates a harmonized functioning of the models across divergent nodes. This configuration epitomizes the foundational setup utilized for the majority of experimental facets. Notably, in line with the recommendations from *Chao et al. (2021)*, the RMSE serves as the primary metric for gauging system efficacy given the employed dataset.

## RESULTS AND DISCUSSION

### System validation

To validate the dynamic updating capability and domain independence of the system, two series of tests were conducted, comparing the proposed system with traditional PdM. The results of these tests are presented in the following section.

### *Dynamic updating validation*

To further assess the dynamic updating of the proposed system, extensive tests were conducted using Root Mean Squared Error (RMSE) as the evaluation metric. Ten rounds of testing were carried out, with each round processing approximately 100k observations of test data through the predictor pipeline. The generated data, along with its corresponding predictions, was integrated into the training pipeline, prompting the model to retrain and adjust to this new data influx. Following this, the predictor pipeline was executed once again with a distinct set of 100k observations, and the results were documented as shown in Fig. 6.

In the time series prediction task, this study thorough tests showed how the dynamic updatable model, the fixed model, and the weighted average approach each have their own roles and benefits. The dynamic updatable model, crafted to adapt continuously and learn from new data, demonstrated exceptional prowess in refining the average RUL predictions.

Given its inherent design, this model's adaptability enables it to capture evolving trends in time series data, thereby delivering more accurate and timely RUL estimates. For instance, during the second round, the dynamic model produced an RMSE of 0.0051, contrasting the fixed model's 0.0067, leading to a conclusive RMSE of 0.0066. Such outcomes affirm the dynamic model's potential to elevate the quality of the final prediction. Comparable enhancements were noticed in rounds 3, 4, 6, and 9. This commendable performance substantiates its utility in real-world settings, where data is in a constant state of flux and models are expected to adjust accordingly.

Conversely, the fixed model offers a consistent foundation for the required predictions. Although it might not perpetually match the dynamic updatable model's performance, its worth is manifested in its unwavering consistency. Initialized during system setup and remaining static thereafter, it furnishes a trustworthy benchmark for predictive endeavors. As an illustration, in the seventh round, the dynamic model's elevated RMSE of 0.0075 was counterbalanced by the fixed model's diminished RMSE of 0.0066, culminating in a final RMSE of 0.0067.

The weighted average methodology serves a pivotal function by amalgamating the predictions of both models. It melds the dynamic updatable model's adaptability with the unwavering nature of the fixed model. The weighted average's parameters can be tailored to either model, thus introducing predictive adaptability contingent on the task's distinct requirements. The parameters $(A)$ and $(B)$ were arbitrarily assigned values, ensuring $(A)$ was no less than 7 and $(B)$ was no more than 3, while the sum is always 1. This strategy ensures prediction stability while simultaneously leveraging the continuous insights furnished by the dynamic updatable model.

### Domain-independent validation

To showcase the domain-agnostic nature of the DL predictive system, experiments were carried out with two unique and independent models. Each model tackled various sets and domains with divergent features and hyperparameters. The initial model utilized a basic setup configuration, while the second was designed around the "measured signals" segment of the dataset, opting for distinct hyper-parameters and substituting LSTM with GRU. Such an approach facilitated a stark contrast in configurations between the two models. The primary goal was to ascertain the system's proficiency in managing diverse data series from different domains while maintaining consistent performance. For performance evaluation, a walk-forward validation method was applied, incorporating a 5-fold cross-validation strategy. Uniform testing data was employed across both domains to guarantee a fair comparison. RMSE served as the chosen evaluation metric to appraise the predictive capabilities of the system.

To represent the results visually, a graph delineating the RMSE values derived from the walk-forward validation for each domain was constructed, as depicted in Fig. 7. The graph shows that the RMSE values are the same across all the folds, which shows that the suggested decentralized DL system works across all domains. As evident from the figure, domain 1 exhibited superior performance across most of the folds, predominantly due to the efficacy of the LSTM-LSTM model configuration. This trend was consistent for folds

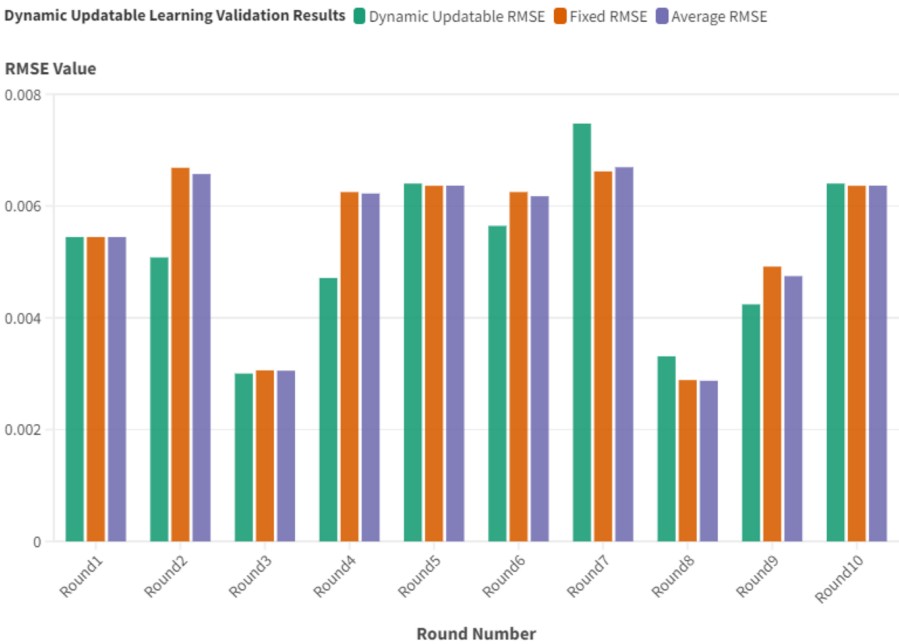

**Figure 6   Dynamic updatable approach validation using basic setup configuration.**

2, 3, 4, and 5. However, fold 1 stood as an exception, with model 2 outpacing model 1 by registering an RMSE of 0.00544 against 0.0061. Notably, the discrepancy between the performances of the two models remained marginal. As a case in point, for fold 3, the RMSE values mirrored each other closely, with only a slight deviation of approximately 0.00043. In conclusion, the walk-forward validation outcomes showcased a consistent RMSE trend across both domains. This strong uniformity proves that the suggested decentralized DL system is domain-independent, showing that it can accurately predict the RUL across a wide range of domains, even when the data has different properties.

## System accuracy

The model was trained using data from units (2, 5, 7, 10, 11, and 15), adhering to the specified initialization in the basic setup, and then tested on units (16, 18, and 20). A Walk-Forward validation strategy was applied, dividing the training dataset into five distinct folds. The insights derived from the experiments on the designed autoencoder model are revealing. By observing the training and validation curves in Fig. 8, it is evident that the suggested model demonstrates an impressive learning rate, with a slight divergence between these curves. Optimal training loss results were achieved at epochs 9 and 12, reaching a low of 0.0019. Regarding validation loss, the lowest value was noted at epoch 20, registering at 0.0019, thanks to the implementation of an early stop approach during the training process. These findings suggest that the suggested model is well balanced, showing

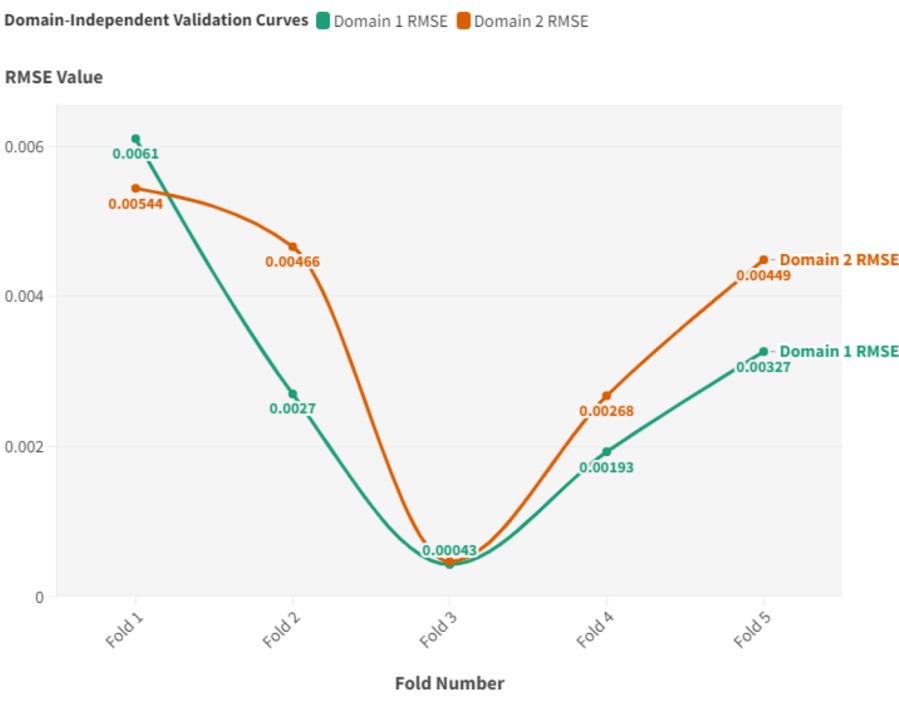

Domain-Independent Validation Curves ■ Domain 1 RMSE ■ Domain 2 RMSE

**Figure 7** **Domain-independent validation curves.**

no signs of overfitting or underfitting, thereby reinforcing its capacity to generalize to new, unseen data.

In comparison with other models based on RMSE, the autoencoder model exhibits superior performance, recording lower error rates, as illustrated in Fig. 9. This reduced RMSE suggests that the suggested model's predictions are more consistently accurate. The LSTM-LSTM model achieved the best performance with an RMSE of 0.0022, reinforcing the choice of this model setup. Other encoder–decoder models also showcased commendable performance. For instance, the GRU-LSTM, GRU-GRU, and CNN GRU models yielded RMSEs of 0.0025, 0.003, and 0.0051, respectively. The effectiveness of encoder–decoder models can be attributed to the context vector, which conveys crucial features from the encoder to the decoder, equipping the decoder with the necessary information for better predictions. A lower RMSE indicates a higher quality of RUL predictions, making the suggested model highly beneficial for precision-demanding applications. In comparison to related studies, this research demonstrates remarkable precision in predictive maintenance, as shown in Table 6. For instance, in the context of RMSE, proposed model yields an impressive value of 0.0022. This stands in stark contrast to the findings presented in the study (*Maulana, Starr & Ompusunggu, 2023*), where the RMSE is reported as 0.05. Furthermore, when compared to *Ma et al. (2021)*, which reports an RMSE of 0.0994, proposed model excels in accuracy. Notably, the article *Berghout et al. (2022)* presents an RMSE value of 5.64, which, when properly scaled, reduces to 0.01 in favorable scenarios. These comparisons highlight the exceptional performance of the proposed predictive maintenance model, underscoring its capacity to enhance reliability

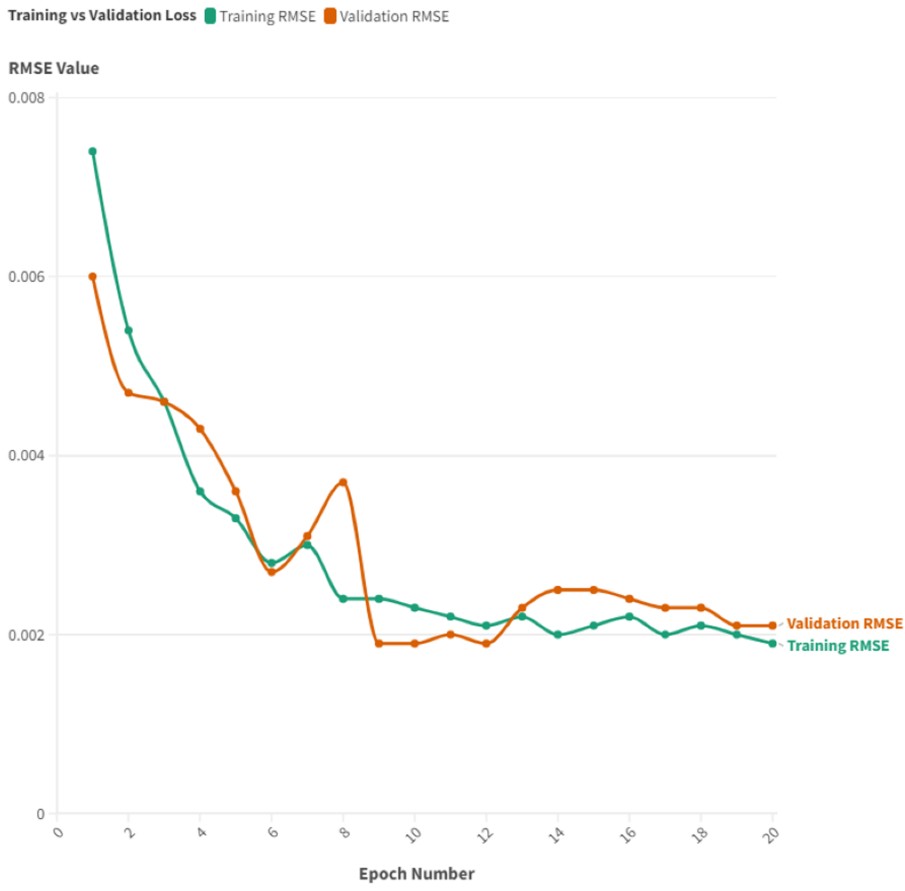

**Figure 8** **Training *vs* validation curve for basic setup configuration.**

and precision in dynamic industrial settings. The encoder-decoder model, purposefully crafted for this system and fine-tuned with dropout techniques, consistently delivers outstanding performance results. This success is further reinforced by the systematic preprocessing methods, affirming the model's reliability and effectiveness.

Our model's performance under varying scenarios of past and future observations, depicted in Fig. 10, offers intriguing insights. The model showcases consistent performance across a wide range of past and future observations, indicating its adaptability. There is a trend where the model's performance peaks with specific combinations of past and future observations, especially when both are below 30. All tests with fewer than 30 past and future observations recorded an RMSE below 0.003. However, the RMSE could rise to 0.0035 for past values nearing 100 observations. This trend suggests the model's proficiency in deciphering inherent patterns in time-series data when considering a specific window of past and future observations.

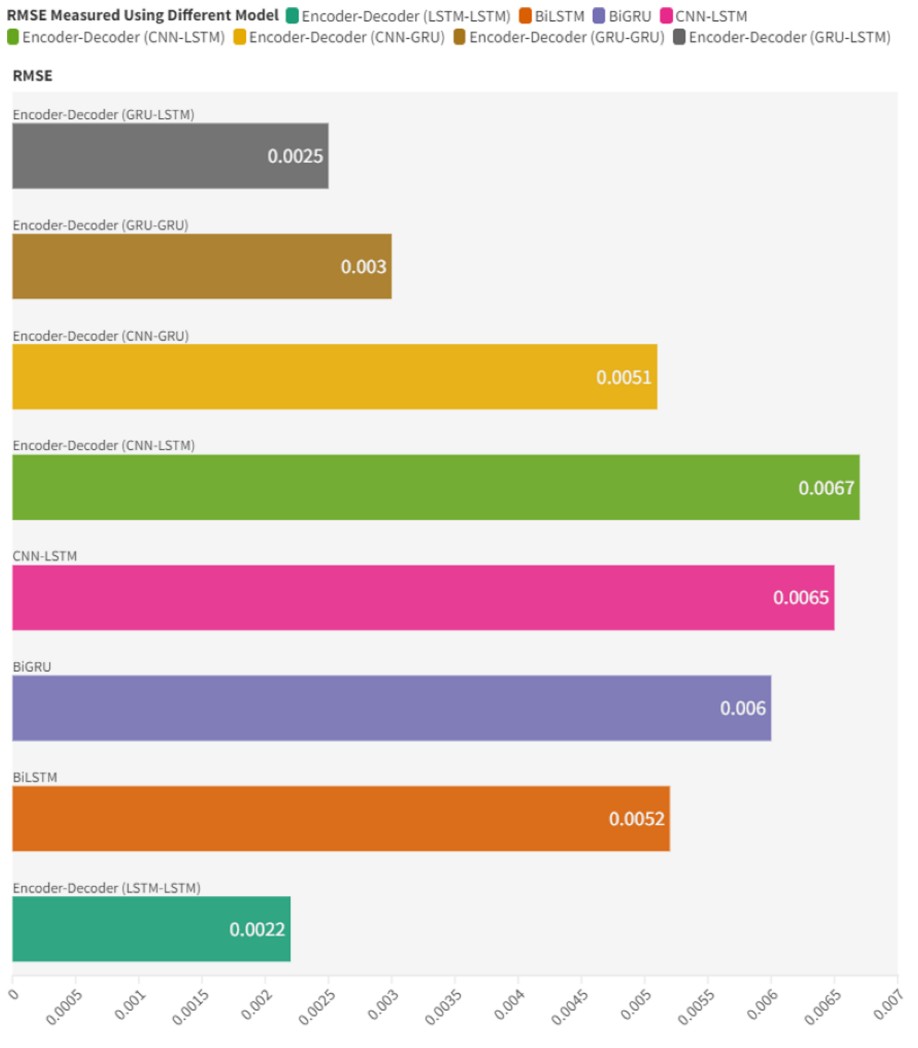

**Figure 9** RMSE results using different DL models for basic setup configuration.

## System performance

Our thorough performance evaluation included three main tests that were meant to see how well and how easily the suggested system could be expanded: the amount of gas used, the time it took to run for two different domain pipelines, and the time it took to run for different amounts of past and future observations. Initially, gas consumption was closely examined during various phases of contract execution, with results presented in Table 7. It is discernible that contract deployment was the primary gas guzzler, with the publish contract alone accounting for a gas usage of 1,362,320 wei. The initialization phase was another notable gas consumer; for instance, the basic setup necessitated a total gas expenditure of $((2*\text{add Node}) + \text{Set Configuration} + (2*\text{Set model}) + \text{Set Training Data} + \text{Change Training Status})$ $((2*25782) + 196584 + (2*145227) + 145184 + 14819 = 753406)$ wei. Intriguingly, post-initialization, the suggested system's gas consumption remained consistently modest, a feat that persisted irrespective of the domain count. Such

**Table 6  Comparison of RMSE values with related studies.**

| Study | RMSE |
| --- | --- |
| Our research | 0.0022 |
| *Maulana, Starr & Ompusunggu (2023)* | 0.05 |
| *Ma et al. (2021)* | 0.0994 |
| *Berghout et al. (2022)* | 0.01 (scaled from 5.64) |

**Figure 10  RMSE given different past (P) and future (F) time steps for basic setup configuration.**

efficiency can be attributed to strategic contract design, which eschewed loops in favor of mapping structures for data storage. As a case in point, the predicting pipeline's gas requirement was $((2 * \text{Get Model Data}) + (2 * \text{Get Configuration}) + \text{Update Train Data})$ $((2 * 24823) + (2 * 30830) + 29222 = 140528)$ wei, while the training pipeline's gas usage was $(\text{Get Train Status} + \text{Update model} + \text{Get Model Data} + \text{Get TrainData} + \text{Get Configuration})$ $(2829 + 31977 + 24823 + 28528 + 30830 = 118510)$ wei. Such efficiency benchmarks underscore the system's scalability and cost-competitiveness.

Our subsequent analysis aimed at deciphering the system's adeptness in multitasking across domains, a metric gauged *via* pipeline execution duration for two contrasting domains. The relevant data is cataloged in Table 8. Intriguingly, the examination revealed minimal incremental increases in execution time, even with the concurrent operation of both domains. Specifically, the cumulative execution duration for the first training pipeline stood at 243.265 s, contrasting with 194.064 s for its second counterpart. On the flip side, the execution duration for the prediction pipelines for the first and second domains was

**Table 7  Execution gas for smart contract aspects.**

| Function | Contracts deployment | Initiation | Predicting pipeline | Training pipeline |
|---|---|---|---|---|
| Get train status | 0 | 0 | 0 | 2,829 |
| Update model | 0 | 0 | 0 | 31,977 |
| Update train data | 0 | 0 | 29,222 | 0 |
| Get model data | 0 | 0 | 24,823 | 24,823 |
| Get train data | 0 | 0 | 0 | 28,528 |
| Get configuration | 0 | 0 | 30,830 | 30,830 |
| Add node | 0 | 25,782 | 0 | 0 |
| Set configuration | 0 | 196,584 | 0 | 0 |
| Set model | 0 | 145,227 | 0 | 0 |
| Set training data | 0 | 145,184 | 0 | 0 |
| Change train status | 0 | 14,819 | 0 | 0 |
| Deploy access contract | 520,152 | 0 | 0 | 0 |
| Deploy configuration contract | 881,819 | 0 | 0 | 0 |
| Deploy publish contract | 1,362,320 | 0 | 0 | 0 |

**Table 8  Execution time for two domains use-case.**

| Aspect | Training pipeline 1(S) | Prediction pipeline 1(S) | Training pipeline 2(S) | Prediction pipeline 2(S) |
|---|---|---|---|---|
| System initialization | 0.074 | 0.067 | 0.079 | 0.072 |
| Blockchain | 0.21 | 0.26 | 0.187 | 0.22 |
| Decentralized storage | 0.205 | 0.31 | 0.152 | 0.286 |
| Deep learning | 242 | 51 | 193 | 42 |
| Data conversion | 0.626 | 0.84 | 0.522 | 0.763 |
| Data preprocessing | 0.15 | 5.23 | 0.124 | 4.97 |
| Total | 243.265 | 57.707 | 194.064 | 48.311 |

57.707 s and 48.311 s, respectively. Such variations in execution times were predominantly dictated by data volume and the hyper-parameters tailored for preprocessing and data transposition.

Lastly, an investigation was conducted to comprehend the relationship between the size of past and future observations and the duration of the training pipeline's execution. The results of this endeavor are graphically represented in Fig. 11. Scrutiny unearthed a proportional relationship between observation sizes and execution times: a surge in the former corresponded to an elongation of the latter. For context, a round encapsulating both the prediction and training pipeline had an execution span of 111 s for 15 past observations juxtaposed with 5 future ones. Yet, this duration spiraled to 996 s when contending with 100 past and 50 future observations. This behavior can predominantly be traced back to the amplified preprocessing and DL computations warranted by an augmented observation window.

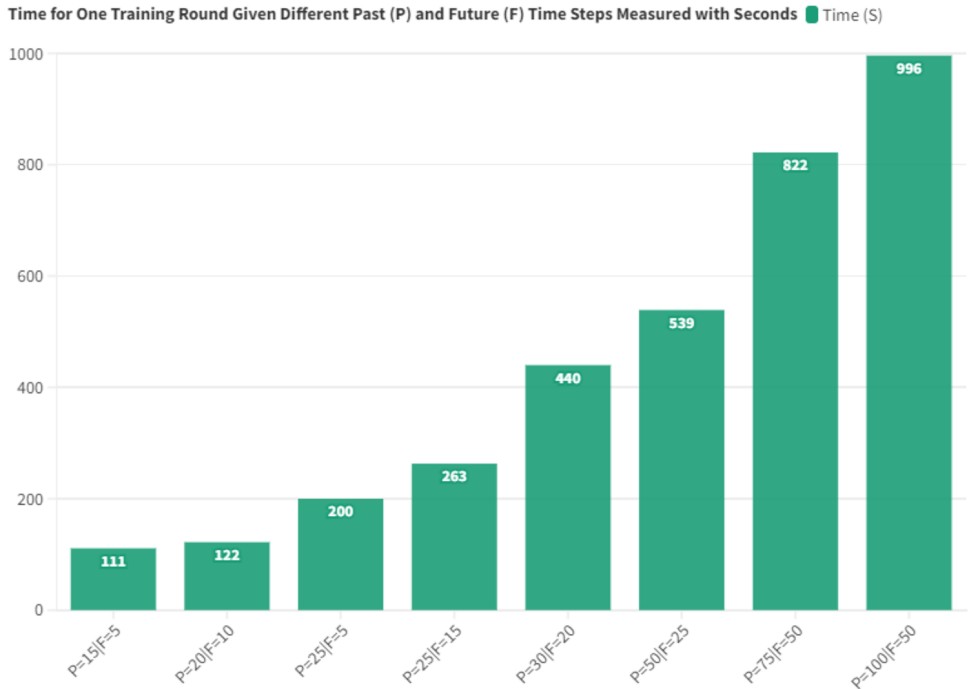

**Figure 11** **Execution time with different past observation (P) and future observation (F).**

## Summary of experimental findings

In summary, the comprehensive validation of the suggested decentralized predictive maintenance system demonstrates its remarkable capabilities. Through extensive testing, we have affirmed its dynamic updating capability, showing how it continuously adapts and learns from new data to improve the accuracy of remaining useful life (RUL) predictions. Additionally, this system exhibits domain independence, successfully managing diverse data series from different domains while consistently delivering reliable RUL predictions. The system's superior performance in terms of RMSE and ability to generalize well to new, untested data highlight its accuracy. Furthermore, it displays adaptability across a range of past and future observation sizes, providing consistent performance. Proposed system's efficiency and scalability are evident in its modest gas consumption and minimal execution time increases, even when multitasking across domains. These findings collectively affirm the robustness and real-world applicability of the suggested decentralized predictive maintenance system, making it a promising solution for dynamic and domain-independent RUL prediction in various industrial settings.

## Security discussion

Considering the attributes of the suggested decentralized DL system, its security can be analyzed based on the CIA triad: confidentiality, integrity, and availability.

### Confidentiality

Encryption and other methods of access control are frequently utilized in the context of blockchain technology and decentralized data storage in order to protect confidentiality resources (*Warkentin & Orgeron, 2020*). The data in the suggested system is spread out across plenty of nodes, which lessens the likelihood of unauthorized access to the information or its publication. In addition, because smart contracts automatically enforce the business logic that is described within them, access controls are maintained by them intrinsically. The purpose of a smart contract, and especially an $(C_a c)$ contract, is to give authentication to every node in the system. This makes it possible for the data to be sourced from authenticated entities. As an illustration, the prediction and training pipeline both make use of a smart contract function that always triggers the check node function in the $(C_a c)$ smart contract.

### Integrity

Integrity is a key component of any system that exchanges or processes data, and it is especially important when those data are used to make important predictions, like the RUL forecasts in the suggested system (*Warkentin & Orgeron, 2020*). Due to a few crucial components, the system's design guarantees a high level of data integrity. Due to its immutable nature, blockchain, which serves as the foundation of the suggested system, delivers great data integrity by default (*Warkentin & Orgeron, 2020*). Data is unchangeable once it is contained in a block and added to the blockchain (*Akhter et al., 2021*).

A permanent and irreversible record is produced in this way. The blockchain ensures the authenticity of the model parameters and training data for the DL models utilized in the system. Accordingly, once a model has been trained and its parameters have been recorded in a block, they cannot be deliberately or unintentionally changed, resulting in predictions that are dependable and consistent. The data integrity of the system is further improved by the use of a decentralized storage solution like IPFS. Data does not change during storage or retrieval because of its distributed nature. Each item of saved data has a distinct hash that confirms its integrity. As a result, the system may be confident that the data it returns for prediction is identical to what it was when it was saved.

Each smart contract in the suggested system $((C_a c), (C_c), and (C_p))$ has a specific purpose, and together they work to maintain the integrity of the addresses of model metadata and training data. The final forecast would not be affected even if an attacker were to successfully inject false data or alter the address of a model or piece of data in the decentralized storage. This is so that the system can confirm the data's origin before processing it. For instance, the outcome of the prediction would be unaffected if attacker $(K)$ is successful in adding model metadata or training data to the decentralized storage. This is because A cannot connect with $(C_c)$ or the predicting contract $(C_p)$ because $(K)$ is not registered by the Access Control Contract $(C_a c)$.

### Availability

The ability of a system to continue to be reachable and operational when its services are needed is referred to as availability (*Warkentin & Orgeron, 2020*). The suggested system's decentralized architecture ensures high availability. The suggested solution makes use

**Table 9 Comparison with the related work the field of PdM (1. Real-time processing, 2. resource limitations, 3. heterogeneity, 4. mobility, 5. scalability, 6. connectivity).**

| Ref | Main techniques | Challenges | | | | | |
|---|---|---|---|---|---|---|---|
| | | 1 | 2 | 3 | 4 | 5 | 6 |
| *Feng & Li (2022)* | ML+DL | ✓ | X | X | X | ✓ | ✓ |
| *Li et al. (2022)* | ML+DL | ✓ | X | X | X | X | X |
| *Ong et al. (2022)* | DL + ML | ✓ | ✓ | ✓ | X | ✓ | X |
| *Bharti & McGibney (2021)* | DL | ✓ | ✓ | X | X | X | ✓ |
| *Lu & Lee (2022)* | ML | ✓ | X | X | X | ✓ | X |
| *Catelani et al. (2021)* | DL | ✓ | X | X | X | X | X |
| *Wu et al. (2021)* | DL | ✓ | X | X | X | X | X |
| *Cheng et al. (2021)* | DL | ✓ | X | X | X | X | X |
| *Ren et al. (2021)* | DL | ✓ | X | ✓ | X | X | X |
| Suggested solution | DL | ✓ | ✓ | X | X | ✓ | ✓ |

of blockchain technology, which allows data to be kept across several dispersed nodes in a distributed ledger rather than on a single central server. This offers two significant advantages. Data is disseminated throughout a network of nodes, preventing a single point of failure in the system. The system can still retrieve the data from another node if a node is rendered inaccessible by unforeseen events like hardware failure, network problems, or other problems. Second, the immutability of the blockchain guarantees that once data has been recorded, it is always accessible, preventing data loss. Decentralized storage enhances availability in a similar way. Due to its decentralized structure, the network nodes only keep the files they are interested in or that users pay them to store. Each file and each block inside it are assigned a distinct fingerprint (*Doan et al., 2022*). A lot of nodes will keep a single file if it is popular; thus, even if some nodes go offline, the file will still be accessible. While the blockchain guarantees the availability of the data, the metadata of the DL models, and the functionality of the smart contracts, the computational tasks related to training and prediction are performed off-chain instead and are thus not constrained by the blockchain's restrictions. As a result, the system's availability is preserved even under tremendous load. The suggested system is incredibly durable and available thanks to the interaction of these many parts, offering dependable service to its consumers.

## Comparative analysis

This research suggest extensive analyses for the above research in comparison with the suggested approach. The analysis focuses on the challenges encountered in the field of PdM, as shown in Table 9. Considering the challenges of PdM, insights are derived from *Alabadi, Habbal & Wei (2022)*, which specifically addresses the challenges associated with IoT in the context of predictive maintenance. As the suggested system relies on data collected from IoT devices, these challenges are directly relevant to the suggested approach. The decentralized system presented effectively tackles these challenges, showcasing its potential to surmount major hurdles encountered in the domain.

Firstly, in terms of real-time processing (*Yasumoto, Yamaguchi & Shigeno, 2016*),the suggested system is designed to handle observations in real time, particularly at the

edge level. This ensures timely and efficient predictions. Secondly, the suggested system addresses resource limitations by distributing the prediction and training processes among separate nodes, thereby reducing resource requirements. Thirdly, scalability concerns are addressed by illustrating the system's capability to manage multiple domains, showcasing its versatility for potential applications across diverse sectors (*Gupta, Christie & Manjula, 2017*). Connectivity is the final issue that the suggested solution addresses. This challenge is partially solved by using blockchain-secure data communication mechanisms, but there is still a need for further improvement regarding the data connectivity between device level and edge level. Acknowledging the need for further improvements in addressing data heterogeneity and mobility challenges is a step towards enhancing the system's robustness and adaptability (*Ghaleb et al., 2016*; *Montori, Bedogni & Bononi, 2016*).

Our system excels at managing time-series data, but heterogeneity is a significant obstacle. Data heterogeneity in the context of PdM may originate from a variety of sources. Diverse varieties of machinery, for instance, will generate distinct sensor readings and operational data, each with its own format and scale. Additionally, data can be collected from multiple locations or under various operational conditions, which adds an additional layer of complexity. Variability in data quality and granularity can be introduced by the use of various types of sensors for data collection. All of these variables contribute to the data heterogeneity. Handling mobility in a decentralized system for PdM based on DL presents several challenges. These include addressing data variability due to changes in equipment location and environmental conditions, managing potential network connectivity issues in remote areas, addressing latency introduced by longer distance data transmissions, and ensuring data security during transmission. For the system to function proficiently in a variety of operational and geographical scenarios, each of these factors requires careful consideration and inventive solutions.

## Limitations and future work

This research, while advancing the field of predictive maintenance, acknowledges certain limitations with plans to address them in future studies:

1. **Blockchain constraints**: Current implementation, while robust, may face challenges in terms of transaction time and processing, particularly when handling large-scale networks.

2. **Dataset limitation**: The proposed solution has been validated for a specific type of dataset. Scaling the solution to accommodate various types of datasets remains a challenge and demands further exploration.

3. **Access list limitations**: As this research presents access list approach, though effective, has its set of constraints, particularly when it comes to scalability in large-scale networks.

4. **Data privacy**: Although we've incorporated decentralized storage, there's still room to enhance data privacy. Current encryption mechanisms can be improved further to safeguard against evolving threats.

5. **Data heterogeneity & system mobility**: Further exploration of data heterogeneity and system mobility challenges is essential to enhance the system's robustness and versatility.

In light of these limitations, future work is set out to explore the following avenues:

- Proposing scalable solutions that can manage larger-scale networks without compromising on the blockchain's inherent advantages.
- Investigating advanced encryption mechanisms to reinforce data privacy in decentralized storage systems.
- Delving into the challenges presented by data heterogeneity and system mobility to ensure the system's seamless functioning across diverse scenarios.
- Considering strategies to optimize the system's performance, especially when handling vast observation sizes, will be pivotal.

## CONCLUSIONS

This study introduces a novel decentralized Predictive Maintenance (PdM) system tailored for accurate Remaining Useful Lifetime (RUL) predictions. By seamlessly integrating blockchain technology, decentralized storage based on IPFS, and deep learning (DL), the proposed system offers innovative solutions to long-standing challenges in traditional PdM methods. In the proposed system, DL handles prediction tasks based on observed data, while blockchain ensures the secure and efficient movement of data. Decentralized storage safeguards crucial model metadata and training data. System in this study is organized into three distinct levels: Device, Edge, and Monitoring levels. It efficiently manages training and prediction pipelines in a decentralized manner, harnessing the coordinated efforts of trainer nodes, predictor nodes, and manager nodes. A notable contribution of this research is the introduction of a dynamic model updating mechanism, a departure from commonly used static models. This mechanism empowers the given system to continuously adapt to evolving conditions within the Industrial Internet of Things (IIoT) environment, ensuring enduring accuracy and practicality in real-world applications. Furthermore, proposed system demonstrates domain-agnostic capabilities, offering the flexibility to handle diverse streams of time-series data. This adaptability eliminates the constraints often associated with traditional PdM approaches, ushering in a new era of flexibility across various industrial domains. The implementation results underscore the feasibility and effectiveness of the proposed system. Lower Root Mean Square Error (RMSE) scores compared to cutting-edge models show that this study achieve superior prediction accuracy. These results validate the dynamic updating and domain independence features of proposed system. Additionally, the performance analysis highlights the system's scalability, even when dealing with varying input and output data scales. Given the paramount importance of security in the IIoT landscape, Suggested system has been fortified with rigorous security measures. A comprehensive analysis encompassing Confidentiality, Integrity, and Availability (CIA) demonstrates the system's robust ability to safeguard both data and entities from potential breaches. Finally, this study provide a transparent overview of the system's limitations and outline directions for future research.

### Funding

The authors received no funding for this work.

### Competing Interests

The authors declare there are no competing interests.

### Author Contributions

- Montdher Alabadi conceived and designed the experiments, performed the experiments, performed the computation work, prepared figures and/or tables, and approved the final draft.
- Adib Habbal supervised the project, prepared the conceptual framework, analyzed the results, authored or reviewed drafts of the article, and approved the final draft.

### Data Availability

The data is available at the NASA Prognostics Center of Excellence Data Set Repository under "17. Turbofan Engine Degradation Simulation-2": https://data.phmsociety.org/nasa/.

The code is available at Zenodo: Montdher Alabadi. (2023). montdher10/Dynamic_DL: Next-generation predictive maintenance: leveraging blockchain and dynamic deep learning in a domain-independent system. (v1.0.0). Zenodo. https://doi.org/10.5281/zenodo.8429718

### Supplemental Information

Supplemental information for this article can be found online at http://dx.doi.org/10.7717/peerj-cs.1712#supplemental-information.

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
