# Peer review of "Next-generation predictive maintenance: leveraging blockchain and dynamic deep learning in a domain-independent system"

_PeerJ Computer Science, doi:10.7717/peerj-cs.1712_

## Round 0.1 · original submission · Major Revisions

One of the reviewers has suggested some citations for your paper, and it is not necessary to cite all of them. You are better to cite those papers that are very related to this work.

Reviewer 1 ·

Basic reporting

No Comment

Experimental design

No Comment

Validity of the findings

The authors should add another section to discuss the validity of their experimental findings.

Additional comments

• The abstract can be improved to follow the structure as Background of the study, objective(s), materials and methods, results, conclusion, and recommendations. At present, it looks like a background introduction.
• The authors should summarize the remaining part of the article in the last paragraph
• Authors should review related works in a new section, state the gaps discovered from the studies reviewed, and note how their work improved on the limitations found in the literature.
• The study should be compared with existing systems (state-of-the-art), and authors should state how it surpassed the existing one and why it performed less or less.
• How did you solve the problem of an overfitting and small dataset
• The limitation of the study should be stated, and they should present future research work.
• Source codes should be provided for replicating the study
• Overall, the English language and presentation style should be improved significantly. There were a lot of grammatical errors and typos. I suggest you have a colleague proficient in English and familiar with the subject matter review your manuscript or contact a professional editing service.
• I have suggested some recent literature from 2023 relating to the study that you are to cite and reference in your article.
a. Lu, S., Liu, M., Yin, L., Yin, Z., Liu, X., Zheng, W.,... Kong, X. (2023). The multi-modal fusion in visual question answering: a review of attention mechanisms. PeerJ Computer Science, 9, e1400. doi: 10.7717/peerj-cs.1400
b. Deng, Y., Lv, J., Huang, D., & Du, S. (2023). Combining the theoretical bound and deep adversarial network for machinery open-set diagnosis transfer. Neurocomputing, 548, 126391. doi: https://doi.org/10.1016/j.neucom.2023.126391
c. Zheng, Y., Li, L., Qian, L., Cheng, B., Hou, W.,... Zhuang, Y. (2023). Sine-SSA-BP Ship Trajectory Prediction Based on Chaotic Mapping Improved Sparrow Search Algorithm. Sensors, 23(2), 704. doi: 10.3390/s23020704
d. Zheng, Y. Y., Zhang, Y., Qian, L., Zhang, X., Diao, S., Liu, X.,... Huang, H. (2023). A lightweight ship target detection model based on improved YOLOv5s algorithm. PLOS ONE, 18(4), e283932. doi: 10.1371/journal.pone.0283932
e. Qu, Z., Zhang, Z., Liu, B., Tiwari, P., Ning, X.,... Muhammad, K. (2023). Quantum detectable Byzantine agreement for distributed data trust management in blockchain. Information Sciences, 637, 118909. doi: https://doi.org/10.1016/j.ins.2023.03.134
f. Wang, S., Sheng, H., Zhang, Y., Yang, D., Shen, J.,... Chen, R. (2023). Blockchain-Empowered Distributed Multi-Camera Multi-Target Tracking in Edge Computing. IEEE Transactions on Industrial Informatics. doi: 10.1109/TII.2023.3261890
g. Song, Y., Xin, R., Chen, P., Zhang, R., Chen, J.,... Zhao, Z. (2023). Identifying performance anomalies in fluctuating cloud environments: A robust correlative-GNN-based explainable approach. Future Generation Computer Systems, 145, 77-86. doi: https://doi.org/10.1016/j.future.2023.03.020
h. Peng, Y., Zhao, Y., & Hu, J. (2023). On The Role of Community Structure in Evolution of Opinion Formation: A New Bounded Confidence Opinion Dynamics. Information Sciences, 621, 672-690. doi: https://doi.org/10.1016/j.ins.2022.11.101
i. Xiong, Z., Li, X., Zhang, X., Deng, M., Xu, F., Zhou, B.,... Zeng, M. (2023). A Comprehensive Confirmation-based Selfish Node Detection Algorithm for Socially Aware Networks. Journal of Signal Processing Systems. doi: 10.1007/s11265-023-01868-6

·

Basic reporting

The strength of the work is in the system structured that has three levels: Device, Edge, and Monitoring levels, enables secure communication with the blockchain and decentralized storage. The flow was demonstrated in the system design and diagrammatically represented. The proposed system aims to deliver a dynamic model updating mechanism that is robust and efficient, but also flexible and adaptable. Irrespective of the strength of the work, there is room for improvement. These is highlighted as follows:
• In the abstract section, the method and result should be clearly stated.
• The authors used possessive pronoun such as ‘our research…, our proposal…’ in the work. I suggest they use phrases like’ this research did …, in the research we propose…
• The article has introduction and background to demonstrate how the work fits into the broader field of knowledge but needs a thorough review to ensure consistency. There should be an addition to the literature to show related work and it has informed this research.
• There should be a clear statement of motivation, leading to this research.
• The structure of the article conformed to the accepted format of ‘standard sections.
• The authors need to organize the statement as a concluding part of the introduction; hence a review of the introduction is needed.
• The materials and methods section is missing and should be added to the work.
• Figure 1 should follow immediately after the writeup about it.
• There should be concluding paragraph in the introduction after the contribution of the work.
• The submission should be ‘self-contained,’ should represent an appropriate ‘unit of publication’, and should include all results relevant to the hypothesis.

• Terms should be confirmed to represent the actual meaning. Example walk forward validation and Rolling Window Validation.

Experimental design

• Materials and Methods should be described as a separate section with sufficient information to enable reproducibility by another investigator.

Validity of the findings

• The finding was not based on subjectivity but from experimental results.
• The data on which the conclusion was made is provided and they are statistically sound
• The conclusion was stated but needs a review to connect to the original question investigated.

---

## Round 0.2 · accepted · Accept

As per comments from the original reviewers, this revised paper can be accepted.

Reviewer 1 ·

Basic reporting

I want to appreciate the effort and dedication you, the authors, have put into revising your manuscript in response to most of my comments. Your revisions have significantly improved the quality and clarity of the paper.

Experimental design

None

Validity of the findings

None

Additional comments

None

·

Basic reporting

The manuscript have been revised and all identified areas for improvement attended to. Therefore, there is improvement.

Literature sources is also improved

Experimental design

The experimental design is meaningful and relevant to the study.

Validity of the findings

The research benefit to literature is clearly stated.